# The neural basis of intelligence in fine-grained cortical topographies

Ma Feilong[1], J Swaroop Guntupalli[2], James V Haxby[1]*

[1]Center for Cognitive Neuroscience, Dartmouth College, Hanover, NH, United States; [2]Vicarious AI, Union City, CA, United States

**Abstract** Intelligent thought is the product of efficient neural information processing, which is embedded in fine-grained, topographically organized population responses and supported by fine-grained patterns of connectivity among cortical fields. Previous work on the neural basis of intelligence, however, has focused on coarse-grained features of brain anatomy and function because cortical topographies are highly idiosyncratic at a finer scale, obscuring individual differences in fine-grained connectivity patterns. We used a computational algorithm, hyperalignment, to resolve these topographic idiosyncrasies and found that predictions of general intelligence based on fine-grained (vertex-by-vertex) connectivity patterns were markedly stronger than predictions based on coarse-grained (region-by-region) patterns. Intelligence was best predicted by fine-grained connectivity in the default and frontoparietal cortical systems, both of which are associated with self-generated thought. Previous work overlooked fine-grained architecture because existing methods could not resolve idiosyncratic topographies, preventing investigation where the keys to the neural basis of intelligence are more likely to be found.

## Introduction

Intelligent thought is the product of efficient neural information processing, but the neural architecture that makes some brains capable of quick wit and deep insight, while others struggle with simple problems, remains an open question. Previous work on the neural basis of intelligence has focused on coarse-grained features of brain anatomy and function, such as the size and shape of the brain and its parts (e.g., *Cox et al., 2019*; *Luders et al., 2007*; *Schmitt et al., 2019*) or the size and strength of connections between large cortical fields (e.g., *Dubois et al., 2018*; *Finn et al., 2015*; *Kong et al., 2019*; *Shen et al., 2017*). Neural information processing, however, is embedded in fine-grained, topographically organized population responses (*Haxby et al., 2014*; *Haxby et al., 2001*). Functional connectivity varies vertex-by-vertex (*Guntupalli et al., 2018*), and even neuron-by-neuron (*Park et al., 2017*), to support such information processing. Therefore, fine-grained functional connectivity depicts in detail how information is exchanged and processed between cortical regions, in contrast to coarse-grained region-by-region connectivity, which depicts the overall synchronization between regions. We investigated whether individual differences in general intelligence are a function of information embedded in fine-grained cortical architecture.

Fine-grained patterns of activity and connectivity can be studied with functional magnetic resonance imaging (fMRI), but the topographies of these patterns are idiosyncratic, impeding study of their role in the neural basis of individual differences in cognitive ability. We used hyperalignment (*Feilong et al., 2018*; *Guntupalli et al., 2018*; *Guntupalli et al., 2016*; *Haxby et al., 2020*; *Haxby et al., 2011*) to resolve the interindividual variation of fine-grained topographies of functional connectivity. Hyperalignment remixes the connectivity profiles of loci in a cortical field into a high-dimensional common space to maximize the similarity across brains in fine-grained, vertex-by-vertex patterns. Individual variations in the residuals around common fine-grained patterns are more reliable than individual variations around common coarse-grained patterns after resolving idiosyncratic

*For correspondence:
james.v.haxby@dartmouth.edu

Competing interests: The authors declare that no competing interests exist.

topographies with hyperalignment (*Feilong et al., 2018*). We found that predictions of general intelligence based on fine-grained (vertex-by-vertex) patterns of connectivity were markedly stronger than predictions based on coarse-grained (region-by-region) patterns. Intelligence was best predicted by fine-grained connectivity in the default and frontoparietal cortical systems, both of which are associated with self-generated thought. These results demonstrate that the neural mechanisms of intelligence reside more in fine-grained interactions of cortical regions than in synchronization of oscillations in large cortical fields.

## Results

We used data from 876 participants of the Human Connectome Project (HCP) (*Van Essen et al., 2013*). Each participant had about 47 min of task fMRI data, collected during the performance of seven tasks, 1 hr of resting-state fMRI data, and scores on 10 cognitive tests. We used connectivity hyperalignment (*Guntupalli et al., 2018*) to factor out idiosyncrasies in fine-grained topographies and model information encoded in local functional connectivity patterns that is shared across brains. We hyperaligned both the HCP task fMRI and resting fMRI datasets. Connectivity hyperalignment projects idiosyncratic cortical topographies for individual brains into a common model connectivity space in which the patterns of connectivity across vertices in a cortical field with connectivity targets elsewhere in the brain are maximally similar across brains. We measured general intelligence (*Spearman, 1904*) as a general factor based on 10 cognitive test scores included in the HCP database (*Dubois et al., 2018*).

For each of the 360 cortical regions, from a parcellation tailored to the HCP dataset (*Glasser et al., 2016*), we computed fine-grained task fMRI and resting fMRI connectivity profiles for each individual (*Figure 1*). The connectivity profile for each cortical vertex differs from profiles for other vertices in the region with a granularity in the common model space that is equivalent to spatial variation across cortical loci within individual brains (*Guntupalli et al., 2018*). We factored out the effect of coarse-grained connectivity by subtracting the mean connectivities between pairs of regions (coarse-grained connectivity profiles; *Figure 1*, middle row) from the vertex-wise connectivities (full fine-grained connectivity profiles; *Figure 1*, top row), to examine the predictive power of fine-grained connectivity unconfounded with prediction based on coarse-grained connectivity. These residual fine-grained connectivity profiles were used throughout the analysis, and we refer to them as fine-grained connectivity profiles for short. Results based on these residual profiles were very similar to those based on full fine-grained connectivity profiles (*Appendix 1—figure 2*). We calculated coarse-grained connectivity on the same hyperaligned data used for calculation of fine-grained connectivity (see *Appendix 1—figure 2*).

We built prediction models based on regional connectivity profiles using cross-validated principal component regression with ridge regularization. We divided the data into *k* test participants from one family (*k* = 1–6) and the remaining 876 - *k* training set participants (*Figure 2*). Thus, each test participant's intelligence was predicted using regression weights derived from other participants' data. We assessed the performance of these models using a coefficient of determination ($R^2$), which denotes the percent of intelligence score variance accounted for (VAF) by the prediction models. We made separate prediction models for hyperaligned fine-scale task and resting connectivity, coarse-grained task and resting connectivity, and non-hyperaligned fine-scale task and resting connectivity.

### Hyperaligned fine-grained versus coarse-grained connectivity profiles

Regional hyperaligned fine-grained task and resting connectivity profiles were highly predictive of general intelligence (*Figure 3A, Figure 4A*) and accounted for twice as much variance in general intelligence compared to coarse-grained connectivity profiles (*Figure 3B, C, Figure 4B, C*).

#### Prediction of intelligence based on task fMRI connectivity profiles

On average across all cortical regions, prediction models based on hyperaligned fine-grained task fMRI connectivity accounted for 27.3% of variance in general intelligence (min: 11.2%; max: 38.9%). In other words, the correlation between predicted and measured intelligence scores ranged from *r* = 0.34 in the least predictive brain region to *r* = 0.62 in the most predictive brain region. By contrast, prediction models based on coarse-grained task fMRI connectivity (*Figure 3B*) accounted on

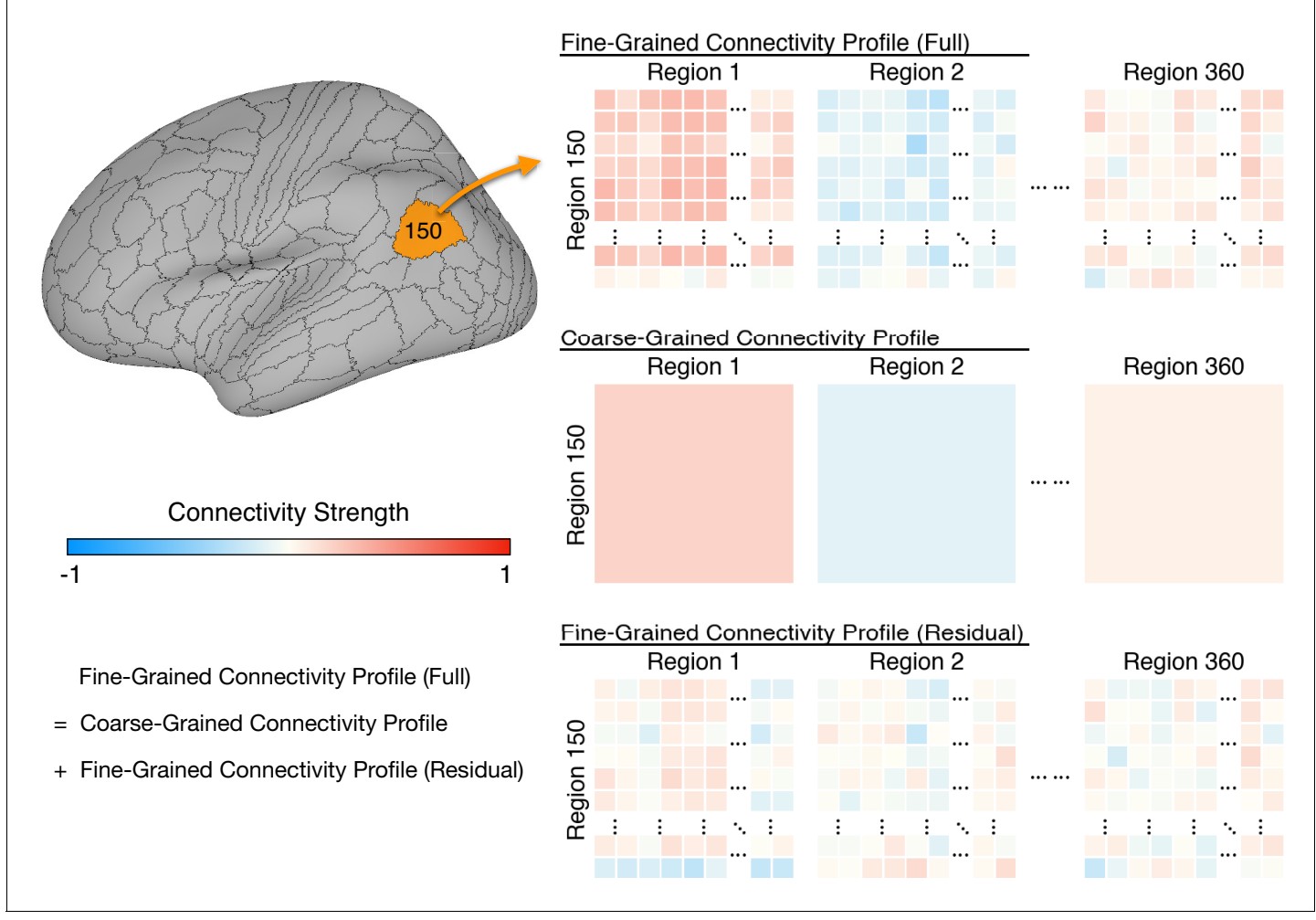

**Figure 1.** Schematic illustration of coarse- and fine-grained functional connectivity. Each brain region (e.g., area 150 shown here) comprises multiple cortical vertices (on average 165). Correlations between their time series and time series for all 59,412 vertices in the whole cortex form that region's full fine-grained connectivity matrix (top row). The fine-grained connectivity profiles for 360 brain regions each have approximately 10 million such correlations. The coarse-grained connectivity profile for the same region (middle row) comprises the average functional connectivity between all of the vertices in that region and all of the vertices in each of the 360 brain regions. Thus, the coarse-grained connectivity profiles for 360 brain regions each have 360 mean correlations. The residual fine-grained connectivity profile (bottom row) for each region is obtained by subtracting the mean correlation for a pair of regions (e.g., regions 1 and 150) from the full fine-grained connectivity profile for that pair and is, thus, unconfounded with coarse-grained functional connectivities. We refer to these unconfounded profiles as fine-grained connectivity profiles for short.

average for only 14.9% of variance (min: 3.3%; max: 26.5%), which is only 54.2% (95% CI: [48.7%, 58.8%]) of VAF by hyperaligned fine-grained connectivity overall. VAF by hyperaligned fine-grained task connectivity was higher than VAF by coarse-grained connectivity in all 360 ROIs (*Figure 3C*). These results suggest that the information encoded in fine-grained interaction patterns between brain regions affords markedly stronger predictions of intelligence than the information in coarse-grained patterns.

The 30 most predictive regions (*Figure 3D, E*; 33.8–38.9% VAF) were in bilateral inferior parietal cortex (15 regions), bilateral medial and superior prefrontal cortex (11 regions), bilateral medial parietal (2 regions), and bilateral posterior lateral temporal cortex (2 regions). The cortices in these regions were predominantly part of the default mode and frontoparietal systems (38.9% and 34.5% of cortical vertices, respectively) with small portions in the ventral and dorsal attention systems (11.3% and 10.6%, respectively; *Figure 3D, E*; *Yeo et al., 2011*). Assignment to cortical systems in a different parcellation tailored to the Glasser parcellation (*Ji et al., 2019*) similarly revealed the dominant role played by the default and frontoparietal systems (*Appendix 1—figure 11*).

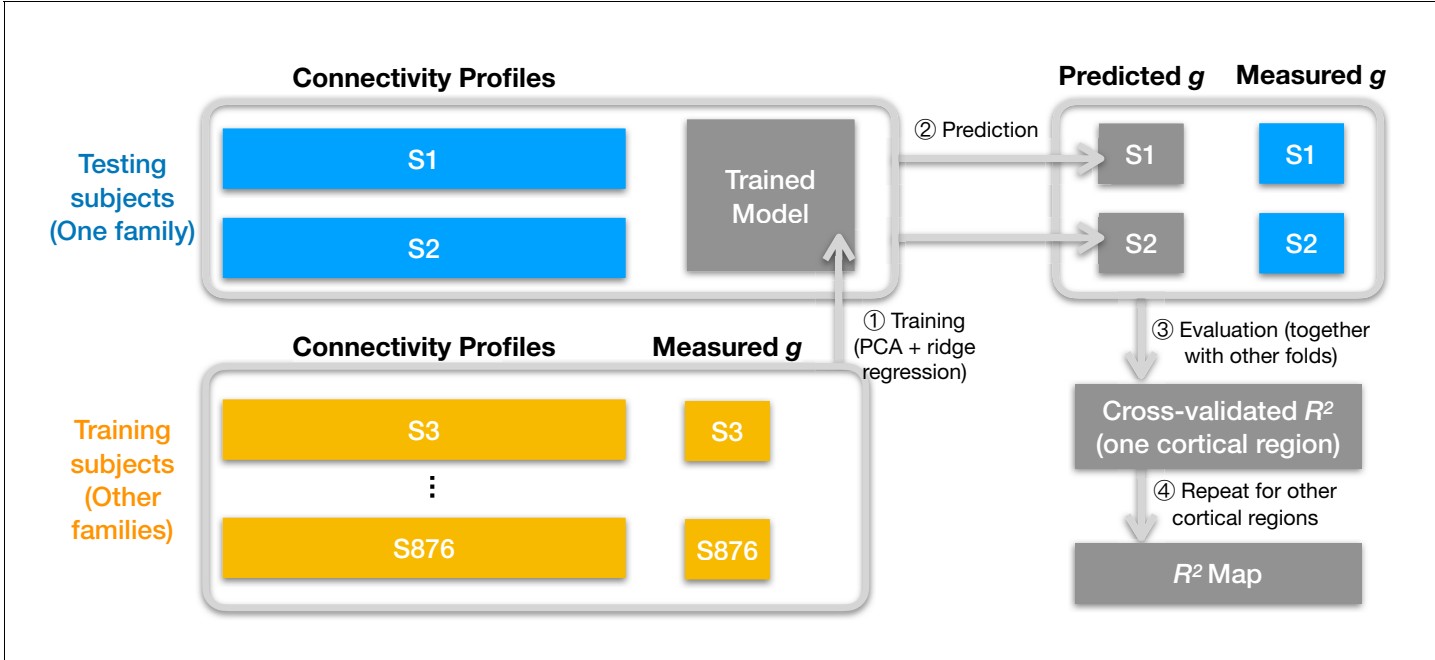

**Figure 2.** Schematic illustration of analysis pipeline. We used a leave-one-family-out cross-validation scheme. For each data fold, we built a prediction model based on the training subjects' data (yellow color) to predict general intelligence ($g$) from the connectivity profile of a cortical region and applied the model to test subjects' connectivity profiles to predict their general intelligence scores (steps 1 and 2). We aggregated predicted and actually measured $g$ across all cross-validation folds to assess model performance with cross-validated $R^2$ (step 3) and repeated this procedure for other cortical regions' connectivity profiles (step 4). After the entire pipeline, we obtained an $R^2$ for each of the 360 cortical regions, which stands for the amount of variance in $g$ that can be accounted for by the region's connectivity profile. We repeated the pipeline for different kinds of connectivity profiles (spatial granularity, dataset, and alignment method) and compared them systematically (*Figure 3, Figure 4, Figure 5*), and these repetitions only differ in the connectivity profiles fed into the pipeline.

## Prediction of intelligence based on resting fMRI connectivity profiles

Prediction of general intelligence based on resting fMRI connectivity showed a similar advantage for hyperaligned fine-grained profiles, relative to coarse-grained profiles (*Figure 4*), but performance was substantially lower than for task fMRI connectivity, consistent with previous reports (*Greene et al., 2018*; *Jiang et al., 2020*). Models based on hyperaligned fine-grained resting fMRI connectivity accounted on average for 19.8% of variance (*Figure 4A*; min: 4.7%; max: 31.2%), whereas models based on coarse-grained connectivity accounted on average for 8.1% of variance (*Figure 4B*; min: −1.2%; max: 17.4%), which is only 40.5% [CI: 34.2%, 45.8%] of VAF by hyperaligned fine-grained resting connectivity. Hyperaligned fine-grained resting fMRI connectivity, compared to coarse-grained resting fMRI connectivity, accounted for more variance in all 360 ROIs (*Figure 4C*). The vertices of 30 regions whose resting hyperaligned fine-grained connectivity was most predictive of general intelligence (25.3–31.2% VAF) had a distribution that was similar to that for hyperaligned task connectivity, sharing 68.3% of cortical vertices (compare *Figure 3D, Figure 4D*), with vertices mostly in the default mode network (49.0%) and smaller parts in the frontoparietal (19.4%), dorsal attention (15.8%), and ventral attention (10.0%) systems (*Figure 4E*; see also *Appendix 1—figure 11*).

Consistent with previous reports (*Greene et al., 2018*; *Jiang et al., 2020*), task fMRI connectivity was significantly more predictive of general intelligence than was resting fMRI connectivity for both hyperaligned fine-grained data (difference = 7.6% VAF; 95% CI: [5.3%, 9.9%]) and coarse-grained data (difference = 6.8% VAF; 95% CI: [5.0%, 8.6%]). Task connectivity accounted for more variance than did resting connectivity in 358 of 360 ROIs for hyperaligned fine-grained data and in 343 of 360 ROIs for coarse-grained data (*Appendix 1—figure 1*).

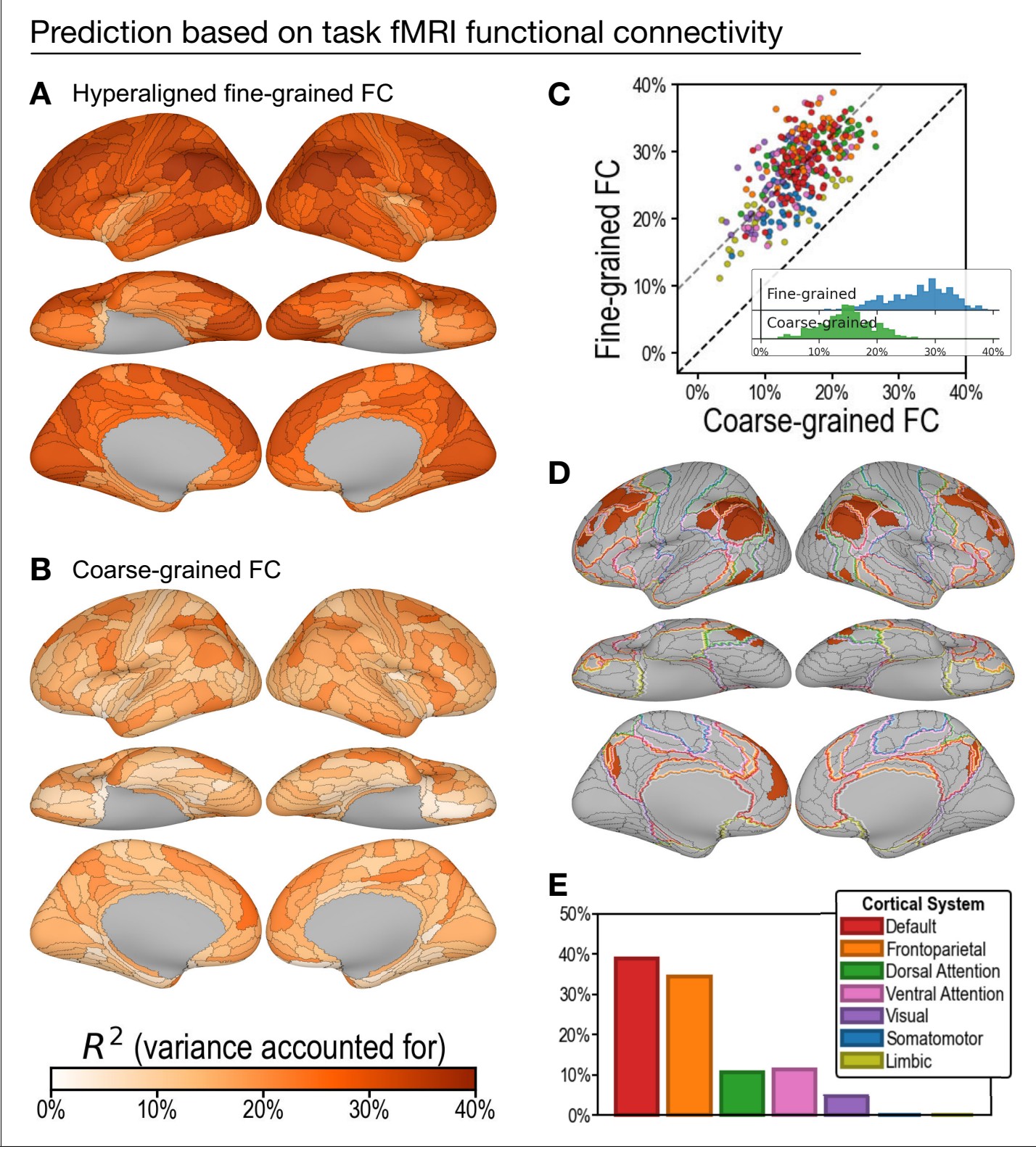

# Prediction based on task fMRI functional connectivity

**A** Hyperaligned fine-grained FC

**B** Coarse-grained FC

$R^2$ (variance accounted for)

0%  10%  20%  30%  40%

**C**

Fine-grained FC / Coarse-grained FC

Fine-grained
Coarse-grained

**D**

**E**

**Cortical System**
- Default
- Frontoparietal
- Dorsal Attention
- Ventral Attention
- Visual
- Somatomotor
- Limbic

**Figure 3.** Predicting general intelligence based on regional task functional magnetic resonance imaging (fMRI) connectivity profiles. Prediction based on hyperaligned fine-grained (A) and coarse-grained (B) connectivity profiles, assessed by the variance in general intelligence accounted for. The scatterplot (C) shows that predictions based on hyperaligned fine-grained profiles accounted for more variance in all 360 regions of interest (ROIs). Prediction models based on each region's fine-grained hyperaligned task functional connectivity profile accounted for 1.85 (95% CI: [1.70, 2.05]) times

*Figure 3 continued on next page*

*Figure 3 continued*

more variance in general intelligence on average than did models based on coarse-grained functional connectivity profiles. Each circle is a cortical region, and the color of each circle corresponds to the cortical system where it resides, using the same color scheme as in (E). Dashed lines denote average difference in $R^2$ (gray) or identical $R^2$ (black). (D) The 30 regions whose hyperaligned fine-grained connectivity best predicted general intelligence are colored as in (A) and (B). The default mode network is outlined in red, and the frontoparietal network is outlined in orange (*Yeo et al., 2011*). (E) Proportion of vertices in these regions that are in seven cortical systems delineated with resting state fMRI functional connectivity (*Yeo et al., 2011*).

## Hyperaligned versus MSM-aligned fine-grained connectivity profiles

To test the efficacy of hyperalignment for revealing predictive individual differences in fine-grained connectivity topographies, we trained another set of models based on fine-grained functional connectivity of data without hyperalignment. These data were aligned with multimodal surface matching (MSM) (*Robinson et al., 2014*). Models based on MSM-aligned fine-grained task connectivity and resting connectivity accounted for, on average across ROIs, 17.6% (min: 5.1%; max: 28.7%) and 11.1% (min: 0.8%; max: 22.5%), respectively, of variance in general intelligence, which was 64.3% (95% CI: [59.9%, 68.3%]) and 56.0% (95% CI: [49.7%, 61.7%]) of that accounted for by models using hyperaligned data (*Figure 5*). Predictions based on hyperaligned fine-grained task connectivity were better than predictions based on MSM-aligned fine-grained connectivity in all 360 regions (*Figure 5C*). Predictions based on hyperaligned fine-grained resting connectivity were better than predictions based on MSM-aligned fine-grained connectivity in 357 of 360 regions (*Figure 5D*). The difference in model performance was larger for brain regions with more functional topographic idiosyncrasy for both task and resting connectivity ($r$ = 0.435 and 0.425, respectively, $p < 10^{-14}$), with two- to threefold increases in VAF in the most idiosyncratic regions. As for hyperaligned fine-scale connectivity and coarse-scale connectivity, MSM-aligned fine-scale task connectivity was more predictive of intelligence than was MSM-aligned fine-scale resting connectivity (difference = 6.5% VAF; 95% CI: [5.0%, 8.0%]; *Appendix 1—figure 1*C). Consistent with our previous findings (*Feilong et al., 2018*), these results show that hyperalignment factors out idiosyncrasies in functional topography to reveal how individuals differ in information encoded in fine-grained cortical functional architecture.

## Discussion

The results show that individual differences in fine-grained patterns of functional connectivity are a markedly better predictor of general intelligence than are coarse-grained patterns, indicating that differences in cortical architecture that underlie inter-individual variation in the efficiency of information processing are more evident at the same spatial scale as topographic patterns that encode information. Discovering the dominant role of fine-grained connectivity patterns in the neural basis of intelligence required a method that resolves individual differences in these idiosyncratic patterns, a method that was not available prior to hyperalignment (*Feilong et al., 2018*; *Guntupalli et al., 2018*; *Guntupalli et al., 2016*; *Haxby et al., 2020*; *Haxby et al., 2011*; *Nastase et al., 2020*).

Differences in intelligence are indexed by tests of the ability to understand and manipulate information embedded in the meanings of words, logical relations, and visual patterns. We predicted that differences in performance would be more related to the fine-grained topographic patterns that represent these kinds of information than to coarse-grained patterns that blur fine distinctions. The results confirm our hypothesis. Hyperaligned, fine-grained, vertex-by-vertex connectivity patterns accounted for 1.85–2.48 times more variance in general intelligence than did coarse-grained, region-to-region connectivity patterns.

To examine the predictive power of fine-grained connectivity unconfounded with coarse-grained connectivity, we built models using the residual fine-grained connectivity profiles. In separate analyses, we built models on the full fine-grained patterns and found that adding coarse-grained information contributed little to VAF (*Appendix 1—figure 2*), underscoring the stronger role that fine-grained cortical architecture plays in the neural basis of intelligence than that played by coarse-grained architecture.

Functional connectivity shows fine-grained variation vertex-by-vertex within a cortical field (*Guntupalli et al., 2018*) and even neuron-by-neuron within an fMRI voxel (*Park et al., 2017*). Transmission of information between cortical fields, therefore, involves more than simple synchronization

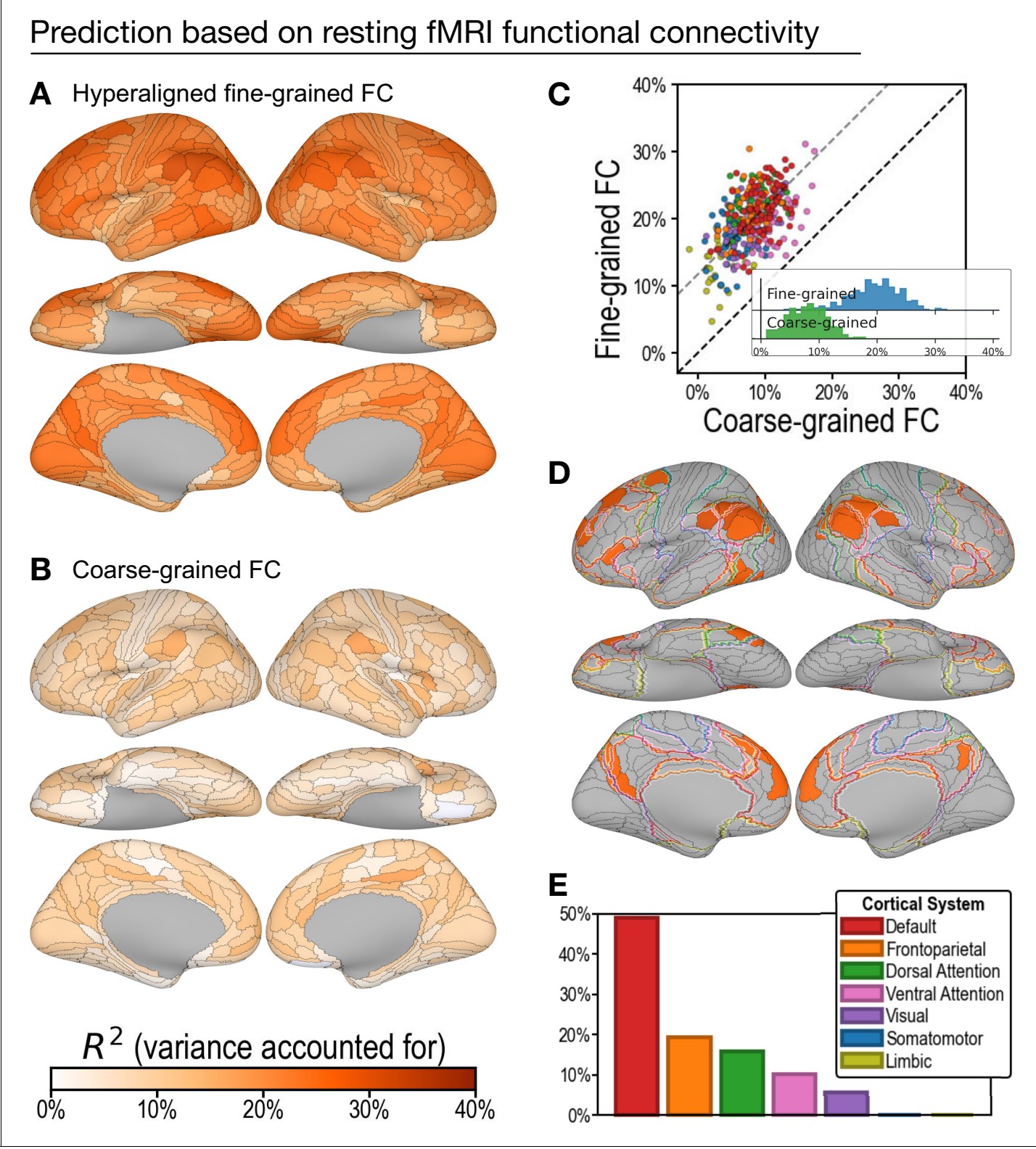

**Figure 4.** Predicting general intelligence based on regional hyperaligned resting functional magnetic resonance imaging (fMRI) connectivity profiles. Prediction based on fine-grained (A) and coarse-grained (B) connectivity profiles. The scatterplot (C) shows that regional predictions based on hyperaligned fine-grained profiles accounted for more variance in all 360 ROIs. Prediction models based on each region's fine-grained hyperaligned resting functional connectivity profile accounted for 2.48 (95% CI: [2.18, 2.93]) times more variance in general intelligence on average than did models

*Figure 4 continued on next page*

*Figure 4 continued*

based on coarse-grained functional connectivity profiles. Each circle is a cortical region, and the color of each circle corresponds to the cortical system where it resides, using the same color scheme as in (E). Dashed lines denote average difference in $R^2$ (gray) or identical $R^2$ (black). (D) The 30 regions whose hyperaligned fine-grained connectivity best predicted general intelligence are colored as in (A) and (B). The default mode network is outlined in red, and the frontoparietal network is outlined in orange (*Yeo et al., 2011*). (E) Proportion of vertices in these regions that are in seven cortical systems delineated with resting state fMRI functional connectivity (*Yeo et al., 2011*).

of global activity. Information is transformed, as evidenced by changes in the representational geometries that are embedded in the fine-grained structure of local topographies (*Connolly et al., 2016*; *Guntupalli et al., 2017*; *Kriegeskorte et al., 2008*). Our results indicate that investigating the efficiency of information processing should focus on fine-grained patterns of connectivity that support information processing.

Hyperalignment resolves functional topographic idiosyncrasies at both coarse and fine spatial scales (see *Jiahui et al., 2020* for an example of aligning functional topographies), which is critical for assessing individual differences in information processing. Alternatively, individualized parcellations (e.g., *Kong et al., 2019*; *Glasser et al., 2016*) can be used to resolve coarse-scale topographic idiosyncrasies, which also improves prediction performance (*Kong et al., 2021*). However, the improvement for coarse-grained functional connectivity was smaller than for fine-grained functional connectivity in predicting general intelligence (*Appendix 1—figure 2*).

Previous work could not reveal the role that information embedded in fine-scale topographies plays in the neural basis of intelligence because individual differences in information were concealed by idiosyncrasies in fine-scale topographies that could not be aligned with prior methods. Hyperalignment resolves the shared information embedded in idiosyncratic topographies, making it possible to investigate these individual differences (*Feilong et al., 2018*). Fine-grained information in hyperaligned data accounted for 1.56–1.79 times more individual variation in intelligence than did fine-grained information in non-hyperaligned data.

Results also show that fMRI data collected during performance of cognitive tasks better predict intelligence than resting state fMRI data, consistent with previous reports (*Greene et al., 2018*; *Jiang et al., 2020*). Previous studies used whole-brain, coarse-grained connectivity for prediction. Here, we show that the added predictive power of task fMRI data, relative to resting fMRI data, extends to both hyperaligned and MSM-aligned fine-grained connectivity patterns.

Our results show further that the neural basis of intelligence resides mostly in the fine-grained structure of connectivity in the default mode and frontoparietal networks (*Buckner et al., 2008*; *Raichle et al., 2001*; *Yeo et al., 2011*). Small parts of the most predictive regions were in the dorsal and ventral attention networks, and an even smaller proportion was in the visual system. The most predictive brain regions did not encroach on the systems for auditory, somatosensory, motor, or limbic function. The predominance of the default mode network was found for both task connectivity and resting connectivity, suggesting that the efficiency of these connections is evident in both. A strong role for the frontoparietal network was more evident in task connectivity, suggesting that the efficiency of this system's connections is better revealed during performance of tasks that increase its activity.

The predominant role of the default and frontoparietal systems in our results suggests that intelligence rests on self-generated and stimulus-independent thinking (*Andrews-Hanna et al., 2014*; *Buckner et al., 2008*; *Dixon et al., 2018*; *Raichle et al., 2001*), or on the apperceptive mass that provides the foundation and context for perception and thought (*Raichle, 2006*), for finding correct solutions to problems and generating novel insights. The functions of the default system are mostly ill-defined because it does not respond to features of external stimuli in a consistent way (*Andrews-Hanna et al., 2014*; *Buckner et al., 2008*; *Golland et al., 2007*). Although default system activity during movie watching and in the resting state is idiosyncratic, it is highly correlated across default areas (*Buckner et al., 2008*; *Golland et al., 2007*) and has a fine-scale structure that can be modeled with hyperalignment (*Feilong et al., 2018*; *Guntupalli et al., 2018*), indicating that this activity has structure and meaning. It has been associated with thinking about others' mental states (*Buckner et al., 2008*; *Frith and Frith, 1999*; *Gobbini et al., 2007*; *Saxe and Kanwisher, 2003*), activation of person knowledge about familiar others (*Gobbini and Haxby, 2007*), autobiographical memory and envisioning the future (*Addis et al., 2007*), as well as other aspects of internally driven

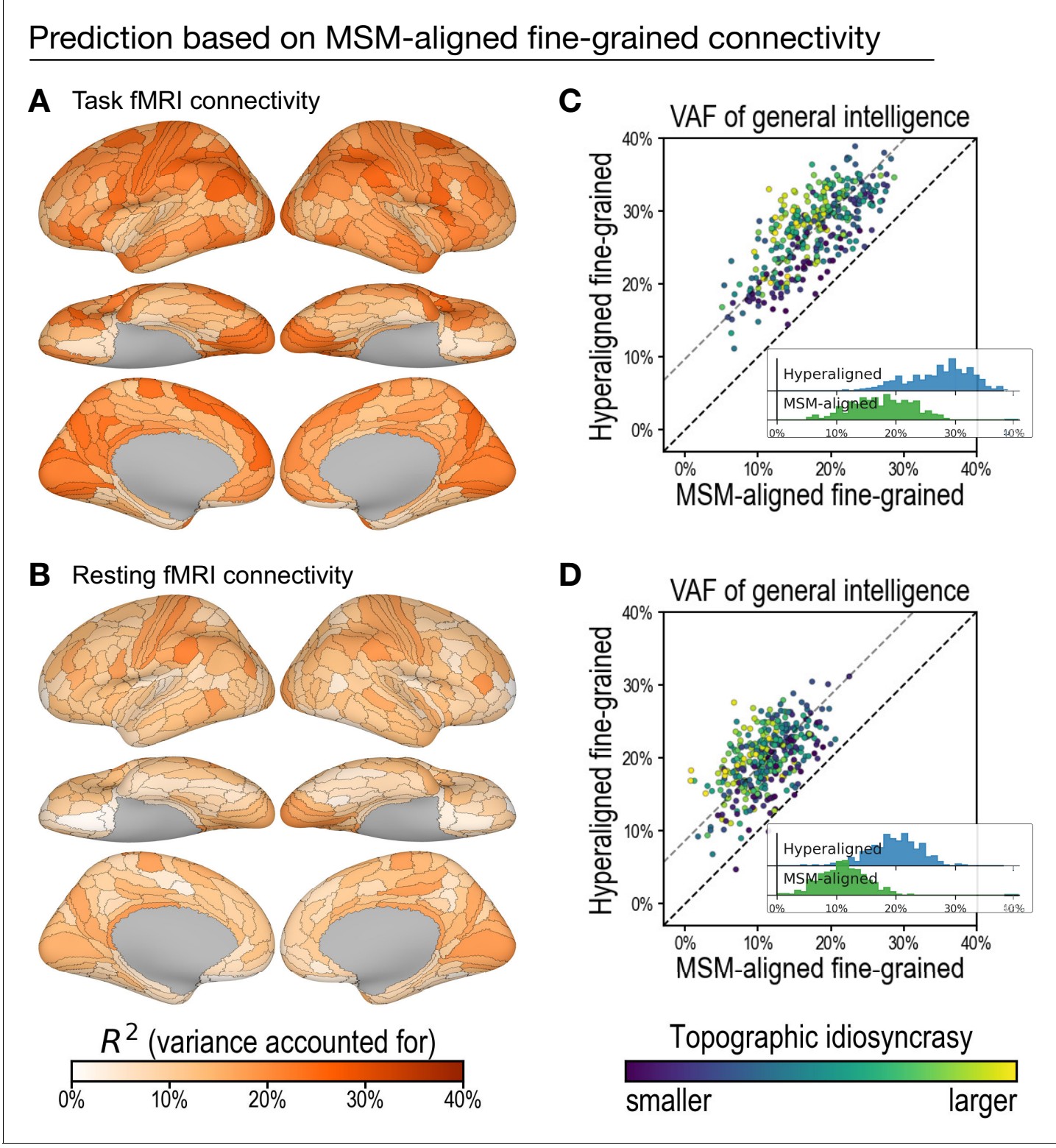

**Figure 5.** Predicting general intelligence based on multimodal surface matching (MSM)-aligned fine-grained connectivity profiles. Prediction based on task functional magnetic resonance imaging (fMRI) (A) and resting fMRI (B) connectivity. Scatterplots compare regional predictions based on MSM-aligned to predictions based on hyperaligned task connectivity (C) and resting connectivity (D) data (comparison to results in *Figure 3A , Figure 4A*).

cognition. The most predictive regions overlapped only a portion of the default network involving primarily the dorsal medial subsystem with smaller parts in the core subsystem (*Andrews-Hanna et al., 2014*; *Andrews-Hanna et al., 2010*). The dorsal medial subsystem has been associated with mentalizing, social cognition, story comprehension, and semantic conceptual processing (*Andrews-Hanna et al., 2014*).

The frontoparietal system may similarly reflect the role of self-directed thought in the neural basis of general intelligence. *Dixon et al., 2018* have identified a subdivision of the frontoparietal network that interacts primarily with the default mode system. They propose that this part of the frontoparietal system is involved in the regulation of introspective processes.

The efficiency of the default system apparently rests more on fine-grained features of its functional connectivity than on coarse-grained functional connectivity. The bases for the organization of its fine-grained topographies and the transformations of representational geometries between default regions are unknown, but the keys to understanding the neural basis of intelligent thought may reside in the information processing embedded in these topographies.

## Materials and methods

### Dataset

We used data from the Human Connectome Project (HCP) (*Van Essen et al., 2013*) for our analysis. The final S1200 data release of the HCP Young Adult dataset contains multimodal MR imaging and behavioral data from 1206 participants that are 22–35 years old. This includes about 1 hr of resting-state fMRI data (four runs, 14.5 min each), 47 min of task fMRI data (seven tasks, two runs each), and extensive cognitive tests for each participant. Out of the 1206 participants, 888 have complete task and resting fMRI data, 1181 have complete scores of 10 cognitive tests, and 876 participants have both. In our analysis, we always used as many participants with complete data as possible to maximize statistical power. That is, we used data from the 1181 participants to derive the general intelligence score, data from the 888 participants to derive hyperalignment models and compute topographic idiosyncrasy, and data from the overlapping 876 participants for prediction analysis.

We used a multimodal cortical parcellation tailored to this dataset (*Glasser et al., 2016*) to delineate cortical regions and compute connectivity targets for hyperalignment as described below. This parcellation is usually labeled as 'Q1-Q6_RelatedParcellation210' or '210P' in the HCP dataset, and sometimes referred to as HCP MMP 1.0.

### Measuring general intelligence

General intelligence is a common factor, often referred to as *g*, underlying all cognitive abilities (*Spearman, 1904*). In this study, we measured general intelligence as the general factor obtained based on 10 cognitive test scores using a factor analysis with a bi-factor model (*Dubois et al., 2018*). These cognitive tests (*Barch et al., 2013*) include those from the NIH Toolbox for Assessment of Neurological and Behavioral Function (http://www.nihtoolbox.org) and additional computerized tests (*Bilker et al., 2012*; *Gur et al., 2010*), covering a range of memory, attention, language, and reasoning abilities. General intelligence can be accurately and consistently identified through different test batteries (*Johnson et al., 2008*; *Johnson et al., 2004*), and our findings are robust over choices of intelligence measures (*Appendix 1—figures 6* and *7*). For this dataset, the general factor of intelligence was derived from the code of *Dubois et al., 2018*, and it accounts for 58% of the covariance structure of cognitive tasks (*Dubois et al., 2018*).

### MRI acquisition

MRI data were acquired with a Siemens 3 T Skyra MRI scanner and a 32-channel head coil at Washington University (*Van Essen et al., 2013*). The scanner was customized with a gradient coil and gradient power. Each subject has 2 T1w and 2 T2w scans with 0.7 mm isotropic voxels, which were used to reconstruct high-resolution cortical surfaces.

Functional MRI data were acquired with a repetition time (TR) of 0.72 s, 2 mm isotropic voxels, and a multi-band acceleration factor of 8. Every volume comprised 72 slices with $208 \times 180$ mm$^2$ field of view each. Half of the fMRI runs were scanned with the LR phase encoding direction, and the

other half were with RL. Spin-echo field maps were collected along with the fMRI data that allow susceptibility distortion correction.

Resting fMRI data were acquired with four runs of 1200 TRs each. Each of the seven tasks were acquired with two runs, and the duration of each run varied across tasks (min: 176 TRs; max: 405 TRs). In total, for each participant, 57.6 min of resting fMRI data and 46.6 min of task fMRI data were acquired. See *Uğurbil et al., 2013* for additional details of MRI acquisition.

## MRI preprocessing

We used the same standard procedure to preprocess fMRI data for all task and resting fMRI data. Our preprocessing was based on the minimally preprocessed version of the dataset (*Glasser et al., 2013*), where data had been corrected for distortion and head motion, and aligned to a standard cortical surface mesh using MSM (*Robinson et al., 2014*), which aims to match multimodal properties across individuals while preserving topology. These data were labeled as 'MSMAll' in the HCP data files, and we refer to them as 'MSM' for short.

We first normalized all fMRI data to percent signal change units, then used linear regression to remove nuisance components from them. These nuisance components include six motion parameters and their derivatives, five principal components from white matter and cerebrospinal fluid (*Behzadi et al., 2007*), global signal, and polynomial trends up to the third order. The white matter and cerebrospinal fluid components were extracted from volumetric data based on eroded anatomical masks. Both the regressors and data were low-pass filtered with a Gaussian kernel (standard deviation = 1 TR) prior to regression. These preprocessing steps were analogous to previous work (*Dubois et al., 2018*; *Finn et al., 2015*; *Greene et al., 2018*; *Shen et al., 2017*).

## Hyperalignment

We performed hyperalignment using these preprocessed data to align the fine-grained cortical functional architecture across individuals and obtained two sets of hyperaligned data (one based on task fMRI data, one based on resting fMRI data) besides the original MSM-aligned dataset. Specifically, we used connectivity hyperalignment (*Guntupalli et al., 2018*) to align fine-grained cortical functional architecture across individuals. We used regional average time series of the 360-region multimodal cortical parcellation (*Glasser et al., 2016*) as connectivity targets. For each of the 59,412 cortical vertices, we computed its connectivity profile as the correlations between its own time series and the 360 regional average time series. Therefore, each connectivity profile is a vector comprising 360 correlation coefficients. In each brain region, we derived a transformation matrix for each individual using these connectivity profiles, which remixes each individual's vertices into a common set of model dimensions through an improper rotation of a high-dimensional space. Each model dimension has similar functional properties across individuals instead of the same anatomical position, and these model dimensions together constitute a common model space. The common model space is usually instantiated as a reference brain, where each vertex corresponds to a model dimension. The information content in each individual's data—the configuration of connectivity vectors in the high-dimensional feature space—is perfectly preserved during the rotation, whereas how and where this information is encoded in idiosyncratic topographies is factored out into the transformation matrix.

In this study, hyperalignment was performed for each brain region separately (i.e., in a similar manner to *Haxby et al., 2011* rather than *Guntupalli et al., 2016*), and the transformation was constrained to be an improper rotation (i.e., rotation that allows reflection) with no scaling. This was to strictly ensure that the information content within a brain region was identical before and after hyperalignment, and only the cortical topography of the information was changed (i.e., better aligned). Searchlight hyperalignment (*Guntupalli et al., 2016*) allows information to be moved in and out of a brain region to some extent, which is preferable in general because the same function does not always reside in the same anatomical region for all participants (*Eickhoff et al., 2018*). However, in this case it can make it difficult to interpret the results. For example, an improvement of prediction performance after searchlight hyperalignment can be caused by better alignment of information in a region, but it can also be caused by additional information moved into the region or noise moved out of the region.

We used the HCP task fMRI dataset to derive the hyperalignment common model and transformations (i.e., to train hyperalignment) when we analyzed the task fMRI data, and resting fMRI

dataset when we analyzed the resting fMRI data. Therefore, the task and resting fMRI datasets remain independent throughout the analysis. It is possible to train hyperalignment using one dataset and apply it to another, and such results are summarized in *Appendix 1—figure 2*.

## Functional connectivity profiles

We used functional connectivity profiles to depict a brain region's functional interactions with the entire cerebral cortex. These connectivity profiles are different from the connectivity profiles used in connectivity hyperalignment as described above.

For each of 360 cortical regions, we computed a fine-grained connectivity matrix for each participant (*Figure 1*, top row), which comprises pairwise interactions between all vertices in the region (165 vertices on average) and all vertices of the cerebral cortex (59,412 vertices). We vectorized this connectivity matrix to form a long vector comprising millions of connectivity strengths (9.8 million on average across all regions), which we refer to as the participant's full fine-grained connectivity profile of the region. This fine-grained connectivity profile depicts in detail the region's information exchange with the rest of the cortex, and it contains information encoded in fine-grained connectivity patterns up to the same resolution as the input fMRI data (2 mm for this dataset).

To separate information encoded in different spatial scales, we split the full fine-grained connectivity profile into a coarse-grained connectivity profile and a residual fine-grained connectivity profile. The coarse-grained connectivity profile for a given region is a vector comprising 360 elements, where each element is the average connectivity strength between all cortical vertices in this region and all vertices in another region (*Figure 1*, middle row). Similar to previous work (*Dubois et al., 2018*; *Finn et al., 2015*; *Shen et al., 2017*), coarse-grained connectivity profiles depict the information exchange between a pair of regions with a single connectivity strength. A residual fine-grained connectivity profile comprises pattern residuals obtained by subtracting the mean connectivity with each region (an element of the coarse-grained connectivity profile) from each element of the vertex-by-vertex connectivity pattern with the region (part of the full fine-grained connectivity profile). Because a coarse-grained connectivity profile comprises region-by-region averages of the corresponding fine-grained connectivity profile, a regression model using coarse-grained connectivity profile (expanded to the same size as the fine-grained connectivity profile) as the independent variable and fine-grained connectivity profile as the dependent variable will always have a slope of 1. Therefore, the difference between the two profiles is also the residual of the regression model. Similar to a full fine-grained connectivity profile, a residual fine-grained connectivity profile comprises millions of elements, and the only difference is that these elements are connectivity residuals instead of connectivities. In other words, a residual fine-grained connectivity profile contains only fine-scale information instead of combined fine- and coarse-scale information.

The analysis involves data aligned in three different ways: MSM, hyperalignment based on task fMRI data, and hyperalignment based on resting fMRI data. For each alignment method, we computed these three kinds of connectivity profiles for each participant and each region. The results of fine-grained connectivity profiles reported in the main text are based on residual fine-grained connectivity profiles, and similar results were obtained using full fine-grained connectivity profiles (*Appendix 1—figure 2*).

## Topographic idiosyncrasy

We measured the level of each region's functional topographic idiosyncrasy as the average dissimilarity of hyperalignment transformation matrices across participant pairs for that region, after correcting for region size. Specifically, for each brain region, we computed the Frobenius norm of the difference between the two transformation matrices for each pair of participants and averaged it across all participant pairs to obtain an average matrix dissimilarity (i.e., average difference matrix norm) for the region. The 360 cortical regions differ in size (measured as the number of vertices in each region). As a result, transformation matrices also differ in size for different regions, and the average matrix norm was predominantly determined by region size. Across all 360 regions, the average matrix norm and the square root of region size had a correlation of $r > 0.9999$. To remove the confounding effects of region size, we fit a linear regression model across the 360 regions using sqrt (region size) as the independent variable and the average matrix dissimilarity as the dependent

variable. We used the residual of the linear regression model to depict the heterogeneity of a region's functional topography across individuals.

We performed hyperalignment twice in our analysis—once using task fMRI data, once using resting fMRI data—and therefore obtained two sets of transformation matrices for each region. Topographic idiosyncrasy indices based on each of the two sets are essentially identical ($r = 0.960$, $p = 4 \times 10^{-200}$). When we analyzed task fMRI data, we used topographic idiosyncrasy based on task fMRI (*Figure 5C*), and when we analyzed resting fMRI data, we used that based on resting fMRI (*Figure 5D*).

## Cross-validation scheme

We used leave-one-family-out cross-validation (*Dubois et al., 2018*) to assess prediction models. The 876 participants were from 411 families, and each family had 2.13 members on average (range: 1–5; with 103 participants with no other family members and 177, 107, 22, and 2 families with 2–5 participants, respectively). Therefore, this cross-validation scheme divided the entire dataset into 411 folds. Each time we leave out a family of $k$ individuals (i.e., related to each other) as test data, and use the remaining 876 - $k$ individuals (i.e., not related to test participants) as training data to train the model. We repeated this procedure, each time using a different family as test data. Therefore, after looping through all families, each individual's general intelligence score was predicted by a model trained with unrelated individuals.

For each cross-validation fold (i.e., each family as test data), we further split the training data into three sub-folds and used nested cross-validation to choose the optimal model parameters. In other words, model parameters were always chosen based only on training data.

To test the robustness of our prediction models against particular ways of splitting data, we replicated our analysis using an alternative cross-validation scheme. Instead of the leave-one-family-out cross-validation, we repeated tenfold cross-validation 50 times, and each time the data was split randomly into 10 chunks in a different way. To accommodate the family structure of the dataset, we ensured that each time individuals from the same family were always in the same chunk. The average model performance across the 50 repetitions was highly similar to that based on leave-one-family-out cross-validation across all regions and connectivity profile types (*Appendix 1—figure 8*), suggesting that our prediction model performance evaluation was accurate and unbiased.

## Regression models

We used principal component regression with ridge regularization to predict general intelligence based on functional connectivity. First, we used principal component analysis (PCA) to derive principal components (PCs) from functional connectivity patterns based on training data. Dimensions in the PC space capture principal ways that individuals' connectivity profiles differ. Thus, the connectivity profile for a region in each training set participant was transformed into a set of scores across PC dimensions. Then, we trained a ridge regression model to predict general intelligence based on these PC scores. The PCA and ridge regression models were then applied to test data to obtain the model predicted general intelligence scores. Model parameters (the number of PCs and the regularization parameter) were always chosen using nested cross-validation as stated above. Candidate model parameters were distributed evenly on a logarithmic scale. Choices for the number of PCs were 10, 20, 40, 80, 160, 320, and all PCs (i.e., no dimensionality reduction). The maximum number of PCs was usually 360 for coarse-grained connectivity profile and 876k for fine-grained connectivity profile. Choices for the regularization parameter α were 81 values from $10^{-20}$ to $10^{20}$.

## Model evaluation

We evaluated our models using the cross-validated coefficient of determination ($R^2$). $R^2$ denotes the percent of variance in general intelligence that was accounted for (VAF) by prediction models. The formula is

$$R^2 = 1 - \frac{\sum_i (y_i - \hat{y}_i)^2}{\sum_i \left(y_i - \bar{y}_{i,\,train}\right)^2}$$

where $y_i$ is the measured score for subject $i$, $\hat{y}_i$ is the model predicted score for subject $i$, and $\bar{y}_{i,train}$ is the prediction by the null model. The prediction of the null model is simply the average score

of all training data, and thus it does not use any information from the features (fine- or coarse-grained functional connectivity in this case) at all. $R^2$ maximizes at 100%, which means perfect prediction; 0 means model performance is only the same as the null model. In rare cases, cross-validated $R^2$ can also be negative, which suggests the model performance is even worse than the null model.

To assess model performance against chance, we used permutation testing to create a null distribution of $R^2$. The HCP dataset has subjects that are from the same family, and they are more likely to have similar general intelligence scores compared with non-related subjects (*Plomin and Deary, 2015*). Therefore, the subject labels are not fully exchangeable, and we used multi-level block permutation (*Winkler et al., 2015*) to resolve the issue. With this hierarchical permutation approach, the data exchangeability is properly modeled, providing more accurate estimates of false positive rates.

For each of the 2160 conditions (360 brain regions × 3 connectivity profile types × 2 fMRI data types), we ran a permutation test by training and evaluating prediction models 100 times based on shuffled subject labels. Each time, we permuted general intelligence scores across the entire dataset in the beginning and re-ran the entire prediction pipeline with these permuted scores. That is, for each cross-validation fold, we chose the optimized parameters using a nested cross-validation based on the training participants and their permuted scores, and used these parameters to train a new model based on the entire training set, which was used to predict the scores of the left-out family. After looping through all cross-validation folds (i.e., all families), we assessed model performance based on the permuted scores. In other words, we repeated the entire process—including parameter optimization, prediction, and model evaluation—using permuted general intelligence score as the target variable instead of the original general intelligence score.

Across conditions (*Appendix 1—figure 4*), the maximum $R^2$ of all 100 permutations was less than the value obtained with the prediction model (i.e., $p < 0.01$) for all models based on hyperaligned fine-grained connectivity or on MSM-aligned fine-grained task connectivity and for all models based on coarse-scale connectivity or on MSM-aligned fine-grained resting connectivity that had an $R^2$ over 1.3%.

## Estimating confidence intervals

We used bootstrap tests to estimate confidence intervals (CIs) for contrasts between VAF by different prediction models. In each of the 1,000,000 repetitions, we randomly sampled a group of 876 individuals used for model evaluation by sampling with replacement from the 876 original individuals. In other words, in each bootstrapped sample, a participant might be selected once, multiple times, or not selected at all. For each bootstrapped sample, we computed VAF differences and VAF ratios. The 95% CI of a difference or ratio is estimated as the 2.5th and 97.5th percentiles of the difference or ratio from the bootstrapped samples (i.e., the sampling distribution of the difference or ratio estimated by bootstrapping participants). The bootstrapping procedure used here only affects the participants used for model evaluation (i.e., the test set), and the model used to predict a participant's score was always trained with the same training set (i.e., participants who are not from the same family as the test participant). This was because prediction performance depends on training sample size (*Appendix 1—figure 8*), and duplicate instances in the training data do not have the same effect on machine learning algorithms as 'real' new data (e.g., noise from two duplicate instances are not independent), which makes the bootstrapped sample no longer representative of the population. Therefore, we only bootstrapped the participants used for model evaluation, so that the sampling distribution is not biased by training sample size or dependency.

## Pearson correlation

We used the Pearson correlation coefficient to assess the relationship between regional topographic idiosyncrasy and the difference in prediction performance based on hyperaligned and MSM-aligned data (*Figure 5*C, D), and the relationship between the square root of a region's size and the average transformation matrix dissimilarity. In both cases, *n* (the number of regions) is always 360, and the degrees of freedom is always 358.

## Dice coefficient

We used the *Dice, 1945* similarity coefficient to quantify the amount of overlap between the 30 most predictive regions based on task fMRI data and those based on resting fMRI data. The formula is

$$DSC = \frac{2|A \cap B|}{|A| + |B|}$$

where *A* is the set of vertices covered by the 30 most predictive regions based on task fMRI data, and *B* is that based on resting fMRI data. |·| denotes set cardinality. *DSC* is short for Dice similarity coefficient.

## Overlap with cortical systems

The cerebral cortex can be divided into seven cortical systems based on functional connectivity (*Yeo et al., 2011*). For each of the 360 regions, we computed the percentage of vertices belonging to each of the seven systems as the number of vertices belonging to the system divided by the total number of vertices in the region. For the 30 most predictive regions, we computed the percentage of vertices across all 30 regions that belonged to each system (*Figure 3E, Figure 4E*). Similar results were found with an alternative division into 12 cortical systems by *Ji et al., 2019* (see *Appendix 1— figure 11*). In the scatterplots (*Figure 3C, Figure 4C*), each region was assigned to one of the cortical systems, which is the system that had the largest amount of vertices in the region.

## Software used

We implemented our analysis using Python and Python packages including NumPy (https://numpy.org/), SciPy (https://www.scipy.org/), and NiBabel (https://nipy.org/nibabel/). The code for performing hyperalignment and nuisance regression was adapted from PyMVPA (http://www.pymvpa.org/) (*Hanke et al., 2009*).

# Acknowledgements

We thank Guo Jiahui, Samuel Nastase, Jeremy Huckins, and Maria Ida Gobbini for helpful comments, suggestions, and discussion. We also thank Yaroslav Halchenko for support with software. Data were provided by the Human Connectome Project, WU-Minn Consortium (principal investigators: David Van Essen and Kamil Ugurbil; 1U54MH091657) funded by the 16 NIH Institutes and Centers that support the NIH Blueprint for Neuroscience Research; and by the McDonnell Center for Systems Neuroscience at Washington University. This work was supported with funds from NSF grants 1607845 and 1835200.

# Additional information

## Funding

| Funder | Grant reference number | Author |
| --- | --- | --- |
| National Science Foundation | 1607845 | James V Haxby |
| National Science Foundation | 1835200 | James V Haxby |

The funders had no role in study design, data collection and interpretation, or the decision to submit the work for publication.

## Author contributions

Ma Feilong, Conceptualization, Software, Formal analysis, Validation, Visualization, Methodology, Writing - original draft, Writing - review and editing; J Swaroop Guntupalli, Conceptualization, Software, Methodology, Writing - original draft, Writing - review and editing; James V Haxby, Conceptualization, Funding acquisition, Visualization, Methodology, Writing - original draft, Project administration, Writing - review and editing

## Author ORCIDs

Ma Feilong https://orcid.org/0000-0002-6838-3971
J Swaroop Guntupalli http://orcid.org/0000-0002-0677-5590
James V Haxby https://orcid.org/0000-0002-6558-3118

## Ethics

Human subjects: Human research participants in the Human Connectome Project gave written informed consent for their participation in accordance with guidelines at participating institutions.

## Decision letter and Author response

Decision letter https://doi.org/10.7554/eLife.64058.sa1
Author response https://doi.org/10.7554/eLife.64058.sa2

# Additional files

## Supplementary files

• Transparent reporting form

## Data availability

The Human Connectome Project data can be downloaded from its database (https://db.humanconnectome.org/data/projects/HCP_1200). Code for the analysis is available at (https://github.com/feilong/IDM_pred copy archived at https://archive.softwareheritage.org/swh:1:rev:ab3f031ab42909774d02f770f4a94d6a2b045eff/.

The following previously published dataset was used:

| Author(s) | Year | Dataset title | Dataset URL | Database and Identifier |
|---|---|---|---|---|
| Van Essen DC, Smith SM, Barch DM, Behrens TEJ, Yacoub E, Ugurbil K, WU-MinnHCP Consortium | 2013 | Human Connectome Project | https://db.humanconnectome.org/data/projects/HCP_1200 | ConnectomeDB, S1200 |

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

# Appendix 1

## Supplemental material
### Comparison of predictions based on task fMRI and resting fMRI data

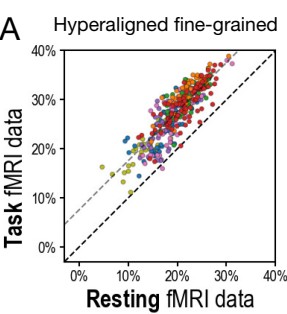 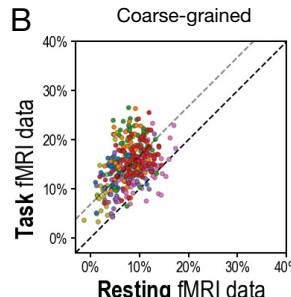 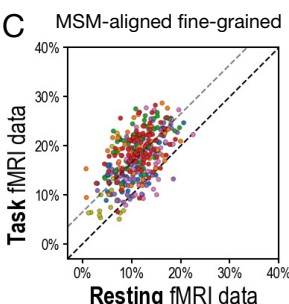

**Appendix 1—figure 1.** Scatterplots comparing variance accounted for by regional prediction models using task functional magnetic resonance imaging (fMRI) and resting fMRI connectivity. (**A**) Hyperaligned, fine-grained connectivity. (**B**) Coarse-grained connectivity. (**C**) Multimodal surface matching-aligned fine-grained connectivity. Each circle represents one region. The color of each circle corresponds to the cortical system that each region is part of (based on a plurality of vertices), using the same color scheme as in *Figure 3D and Figure 4D*.

Results showed that task fMRI data produced better predictions of general intelligence than did resting fMRI for hyperaligned residual fine-grained, coarse-grained, and MSM-aligned residual fine-grained connectivity.

## Comparison of alignments and predictions based on single task connectivity

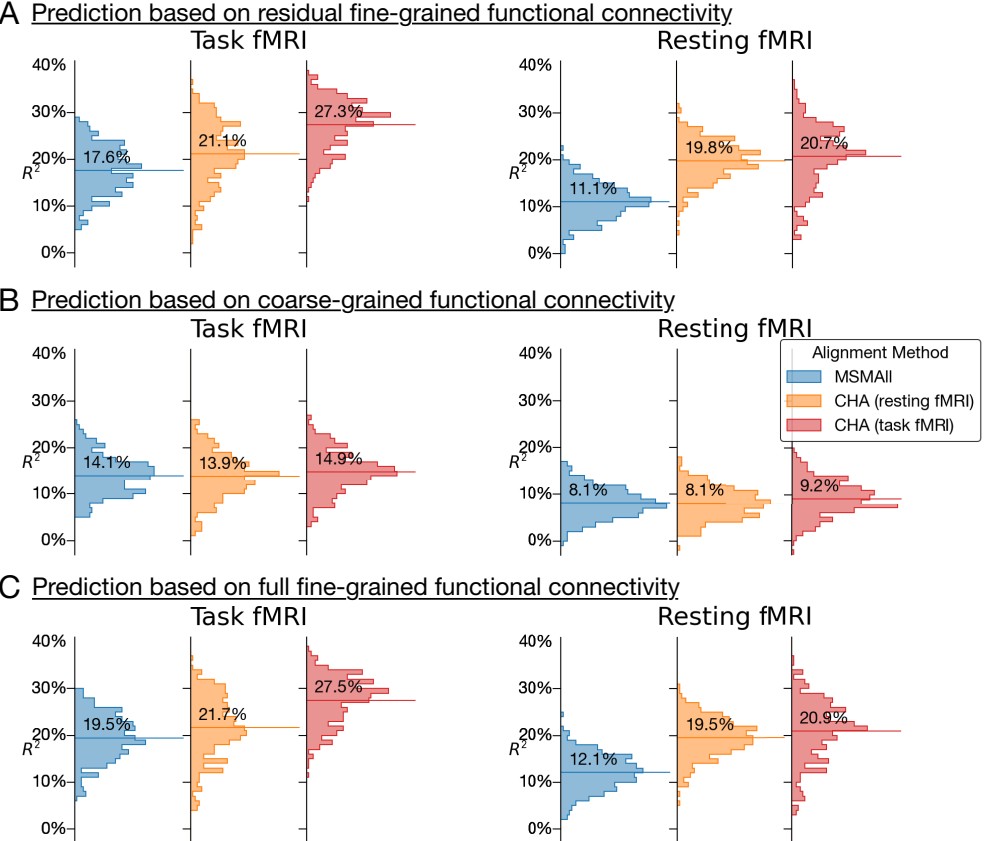

**Appendix 1—figure 2.** Histograms of variance accounted for by predictions of general intelligence using different functional magnetic resonance imaging connectivity datasets and different alignment methods. (**A**) Prediction based on residual fine-grained connectivity. (**B**) Predictions based on coarse-grained connectivity. (**C**) Predictions based on full fine-grained connectivity, which includes information in both fine-grained and coarse-grained patterns. CHA: connectivity hyperalignment.

Connectivity hyperalignment based on task fMRI data produced stronger predictions of general intelligence based on task connectivity than did connectivity hyperalignment based on resting fMRI data. Predictions based on resting connectivity were equivalent for connectivity hyperalignment based on task fMRI and resting fMRI. Prediction based on hyperaligned residual fine-grained connectivity and on hyperaligned full fine-grained connectivity profiles (A and C) were essentially identical.

Hyperalignment transformations remix the time series for cortical vertices, which alters the mean of correlations between the vertices of two regions, namely, the coarse-scale functional connectivity between these regions (B). Analyses of hyperaligned fine-grained connectivity profiles all used the coarse-grained connectivity profiles calculated on the same hyperaligned data for comparison. In practice, the effect of calculating coarse-grained connectivity profiles on task hyperaligned task fMRI, hyperaligned resting fMRI, and MSM-aligned fMRI is small and does not affect the results of these analyses (*Appendix 1—figure 2*).

## Predicting intelligence based on connectivity during single tasks

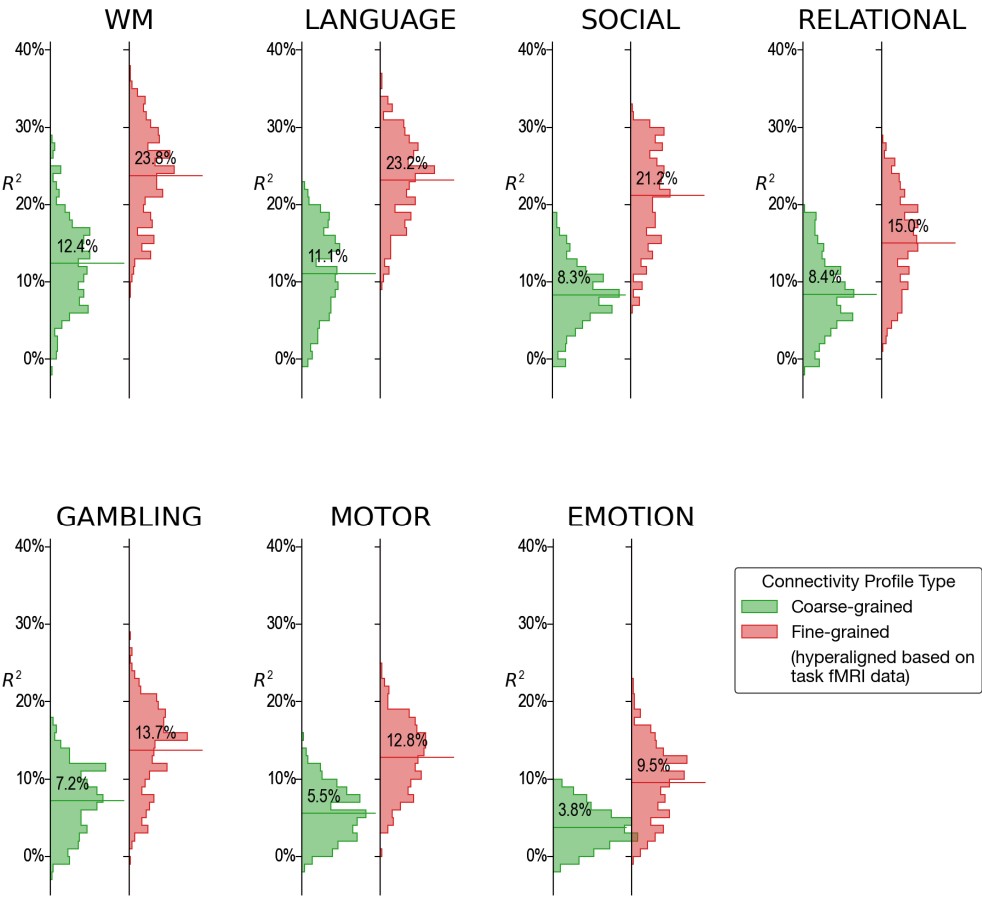

**Appendix 1—figure 3.** Histograms of variance accounted for by regional prediction models based on connectivity calculated from functional magnetic resonance imaging (fMRI) during single tasks. For every single fMRI task (4.2–9.7 min), fine-scale hyperaligned connectivity data produced better predictions of general intelligence than did coarse-scale connectivity. The strongest predictions from single task fMRI were obtained from data during performance of the working memory (WM), language, and social cognition tasks.

## Null distribution of model performance based on permutation testing

**A** Permutation results based on task fMRI data

**B** Permutation results based on resting fMRI data

**Appendix 1—figure 4.** Permutation testing. For each of the 2160 conditions (360 brain regions × 3 connectivity profile types × 2 functional magnetic resonance imaging data types), we ran a permutation test by training and evaluating prediction models 100 times based on shuffled subject labels. (A) and (B) show results based on task fMRI data and resting fMRI data, respectively. For 2148 out of 2160 conditions (99.4%), the actual $R^2$ was larger than all 100 permuted $R^2$s. For the remaining 12 conditions, the actual $R^2$ was close to 0 (all <1.3%). In other words, for any model that had an $R^2$ of at least 1.3%, it is also larger than the maximum $R^2$ of all 100 permutations (i.e., $p < 0.01$).

Distribution of model parameter choices based on nested cross-validation

A Parameters of models trained on task fMRI data

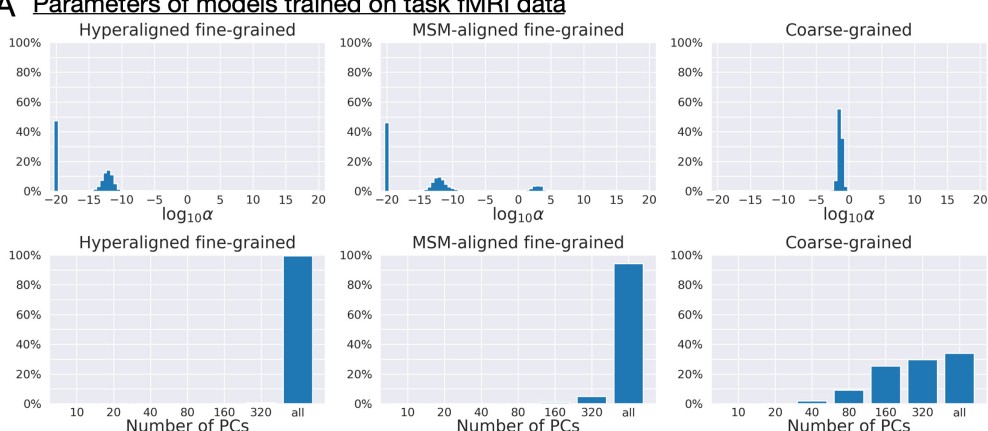

B Parameters of models trained on resting fMRI data

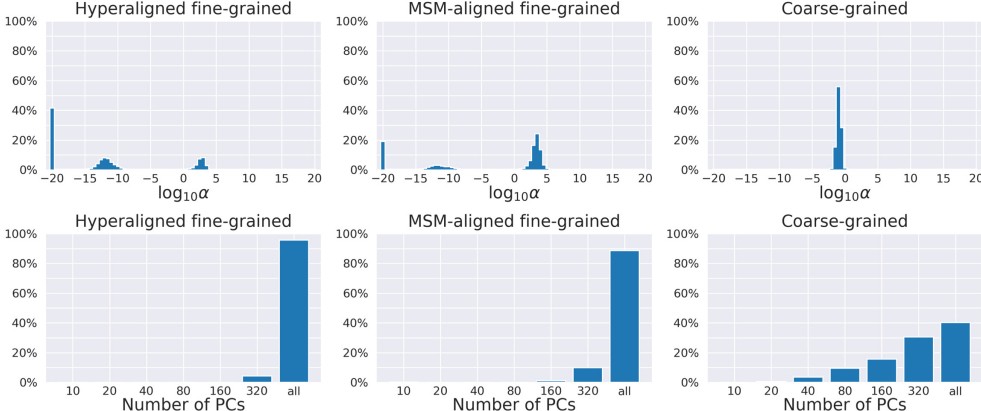

**Appendix 1—figure 5.** Distribution of regularization parameters $\alpha$ and number of principal components (PCs) for prediction models. For each kind of functional magnetic resonance imaging (fMRI) data (A: task fMRI; B: resting fMRI) and each kind of connectivity profile (hyperaligned fine-grained, multimodal surface matching-aligned fine-grained, coarse-grained), we summarize the distribution of model parameters (the regularization parameter, $\alpha$, and the number of PCs) across all cross-validation folds (411 families × 360 regions = 147,960). The maximum number of PCs was usually 360 for coarse-grained connectivity profile and close to 876 for fine-grained connectivity profile (depending on training sample size). Models trained on fine-grained connectivity profiles (left and middle columns) tend to use less regularization (smaller $\alpha$s) and more PCs compared with models trained on coarse-grained connectivity profiles (right column), especially models trained on hyperaligned fine-grained connectivity profiles. More PCs used by the model suggest that there are more dimensions in the connectivity profiles that are related to general intelligence, and less regularization suggests these PCs contain more signal relative to noise.

# Predicting alternative measures of intelligence

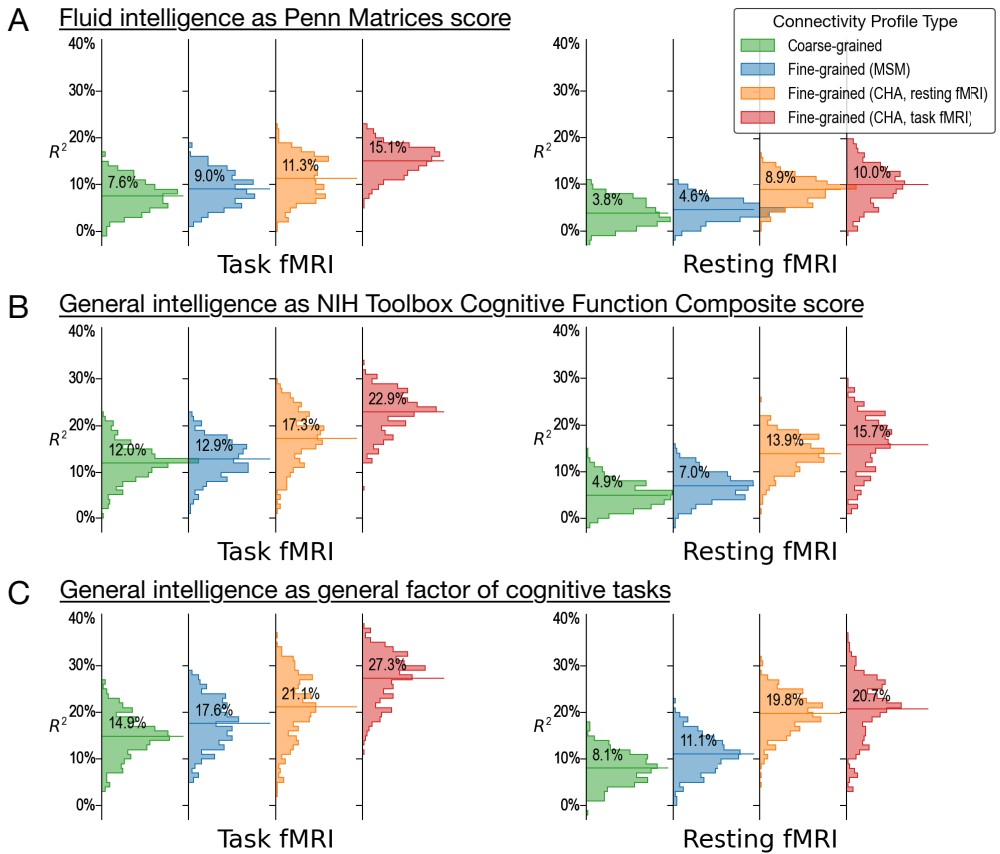

**Appendix 1—figure 6.** Predicting alternative measures of intelligence based on connectivity profiles. Our prediction models and connectivity profiles can also be trained to predict other measures of intelligence, such as fluid intelligence measured with the Penn Matrices (**A**, variable 'PMAT24_A_CR' in the dataset) and general intelligence measured as the cognitive function composite score of the NIH Toolbox (**B**, 'CogTotalComp_Unadj'). Similar to the results of the main paper (shown here as **C** for comparisons), predictions based on fine-grained hyperaligned connectivity profiles account for approximately two times more variance in intelligence compared with coarse-grained connectivity profiles for task functional magnetic resonance imaging (fMRI) data, and three times more for resting fMRI data. The four columns on the left side are based on connectivity profiles computed from task fMRI data, and the four on the right from resting fMRI data. For each kind of data, we computed fine-grained connectivity profiles based on three different alignment methods (multimodal surface matching, hyperalignment based on resting fMRI data, and hyperalignment based on task fMRI data), which we colored as blue, orange, and red, respectively. The orange and red colors denote the kind of data used to derive hyperalignment transformations. In the results shown in this figure, they may not be the same as data used to compute connectivity profiles.

Fluid intelligence as Penn Matrices score

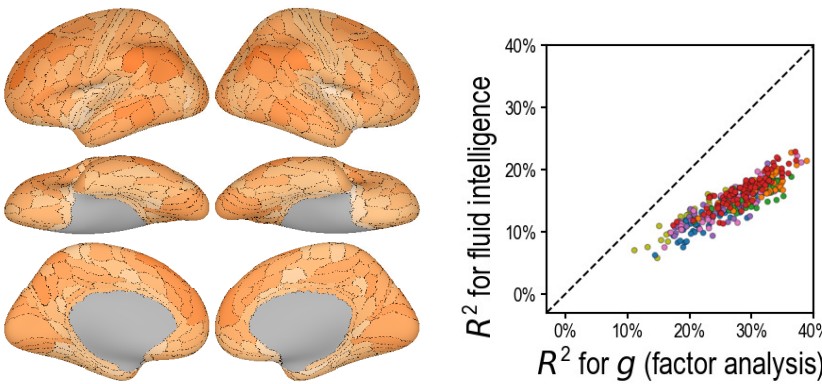

General intelligence as NIH Toolbox Cognitive Function Composite score

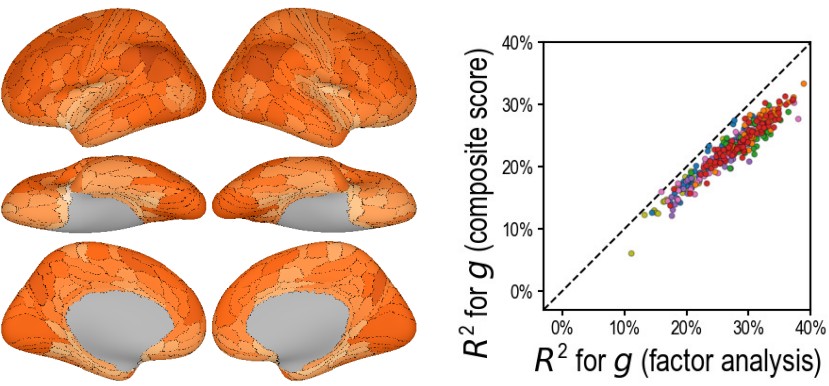

**Appendix 1—figure 7.** Predicting alternative intelligence measures based on regional hyperaligned fine-grained task functional magnetic resonance imaging connectivity profiles. Regional connectivity profiles that were more predictive of factor analysis-based general intelligence scores were more likely to be more predictive of matrix reasoning-based fluid intelligence scores (top row, *r* = 0.92) and NIH Toolbox-based general intelligence scores (bottom row, *r* = 0.96). This suggests that our results were robust over intelligence measure choices.

Alternative cross-validation schemes and the effect of training sample size

### Task fMRI functional connectivity

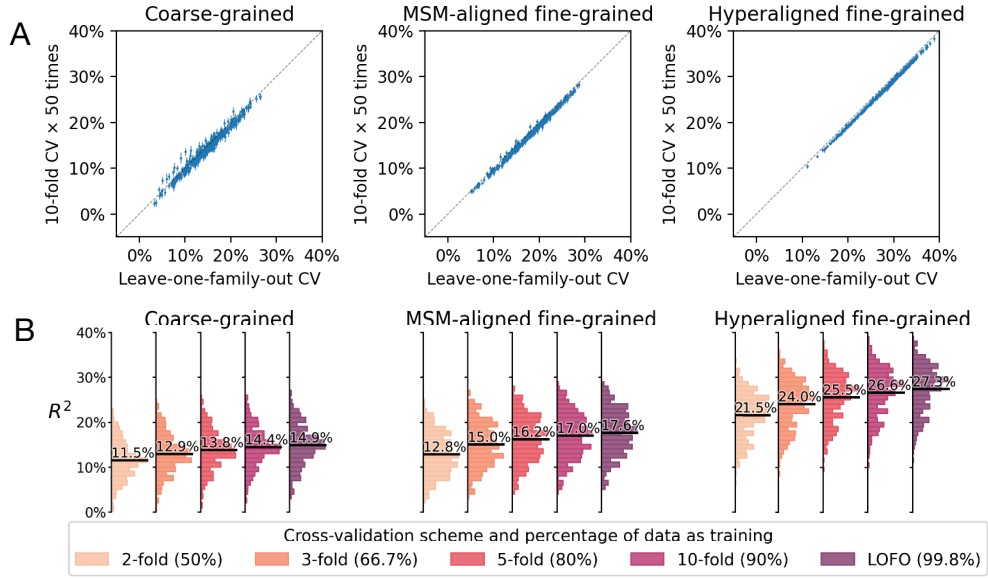

### Resting fMRI functional connectivity

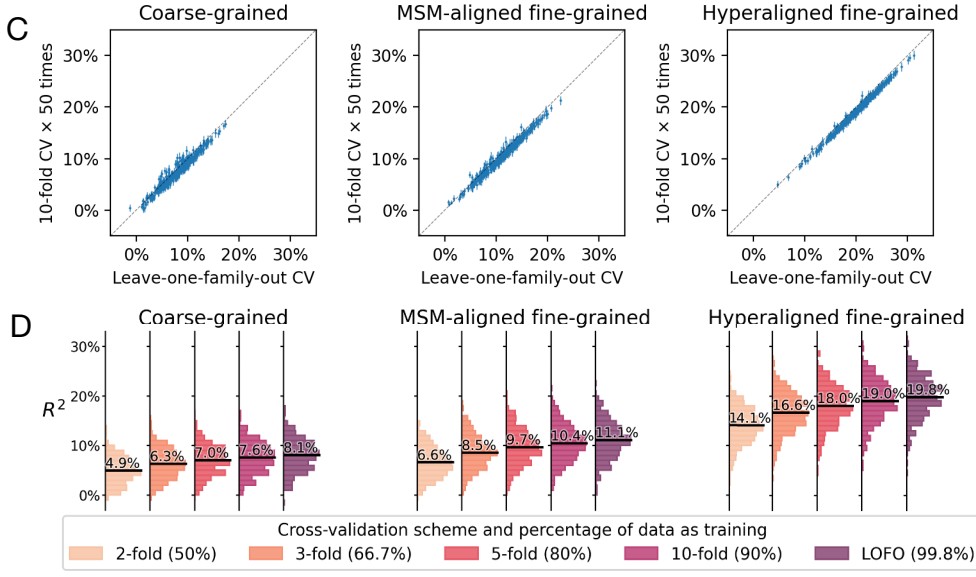

**Appendix 1—figure 8.** Comparison between cross-validation schemes. We replicated our analysis using the tenfold cross-validation scheme (repeated for 50 times) and compared model performance based on tenfold cross-validation (average across 50 repetitions) with that based on leave-one-family-out cross-validation (**A**, **C**). Each dot denotes a brain region, and error bars denote the standard deviation of $R^2$ across the 50 repetitions. All dots were close to the identity line (gray dotted line), which indicates that model performance was highly similar for leave-one-family-out cross-validation and tenfold cross-validation. On average across all regions, model performance based on leave-one-family-out cross-validation was slightly higher than that based on tenfold cross-validation, which was likely driven by the difference in training sample size (i.e., approximately 90% versus 99.8% of the entire dataset). This was confirmed by additional analysis showing further reduction of model performance based on smaller k values and correspondingly smaller training

*Appendix 1—figure 8 continued on next page*

sample sizes (**B**, **D**; k = 2, 3, or 5 for k-fold cross-validation; training data was 50%, 66.7%, and 80% of the entire dataset, respectively).

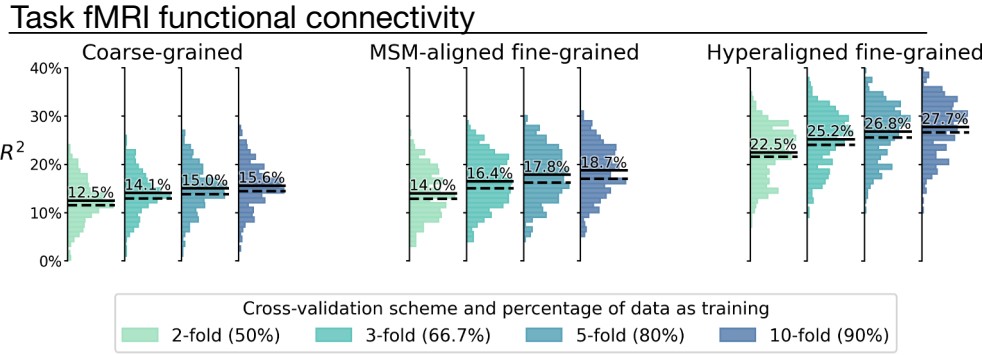

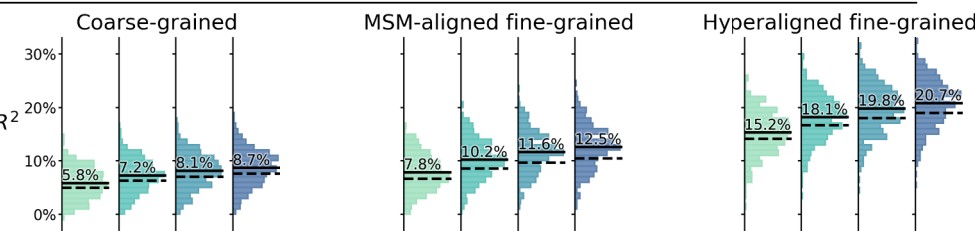

**Appendix 1—figure 9.** Overestimation of model performance due to lack of data independence. We trained new prediction models based on the k-fold cross-validation scheme, but without controlling for family structure. That is, for each testing participant, other members from the same family might be in training data. Compared with the k-fold results that controlled for family structure (*Appendix 1—figure 8*; dashed lines in this figure), results without controlling for family structure consistently overestimate model performance (average $R^2$ difference across regions: 0.9–2.1%; solid lines in this figure). This demonstrates the necessity of ensuring data independence between training and testing data to avoid biased model evaluations (see *Varoquaux et al., 2017* for a similar issue with leave-one-trial-out). Specifically, training and testing data should not have members from the same family.

## The effect of hyperparameter choices on model performance

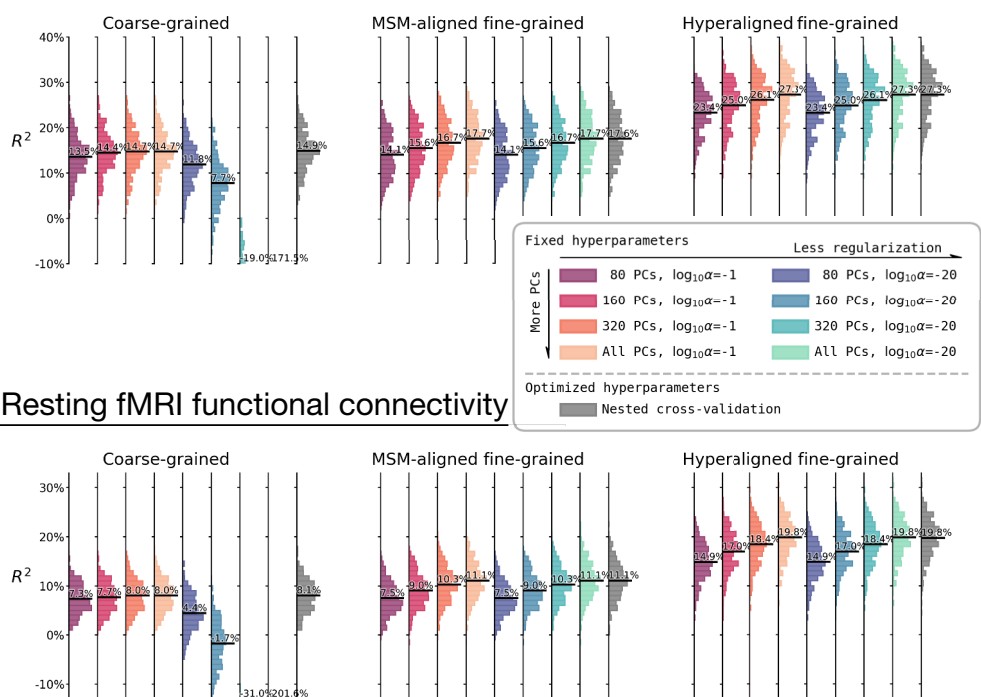

**Appendix 1—figure 10.** The effect of hyperparameter choices on prediction performance. Besides using fine-tuned hyperparameters based on nested cross-validation (rightmost columns in gray), we trained prediction models based on another eight sets of hyperparameter choices. These eight sets of hyperparameters are combinations of four levels of dimensionality reduction (80 principal components [PCs], 160 PCs, 320 PCs, or all PCs) and two levels of regularization ($\alpha$ = 0.1 or $\alpha$ = $10^{-20}$). These hyperparameter levels are the levels most frequently chosen based on nested cross-validation (*Appendix 1—figure 5*). The histograms denote $R^2$ distribution across brain regions, and horizontal bars are the average $R^2$ across regions. For prediction models based on coarse-grained connectivity profiles, regularization is critical for prediction model performance, and with insufficient regularization models overfit dramatically with more PCs. When the regularization is large enough, the model performance slightly increases with higher number of PCs. For prediction models based on fine-grained connectivity profiles, model performance is hardly affected by regularization level and consistently increases with higher number of PCs. This suggests that PCs based on fine-grained connectivity profiles contain more information related to individual differences in intelligence and less noise. Note that even with only 80 PCs prediction models based on hyperaligned fine-grained connectivity profiles still account for approximately two times more variance than those based on coarse-grained connectivity profiles.

## Alternative definition of cortical networks

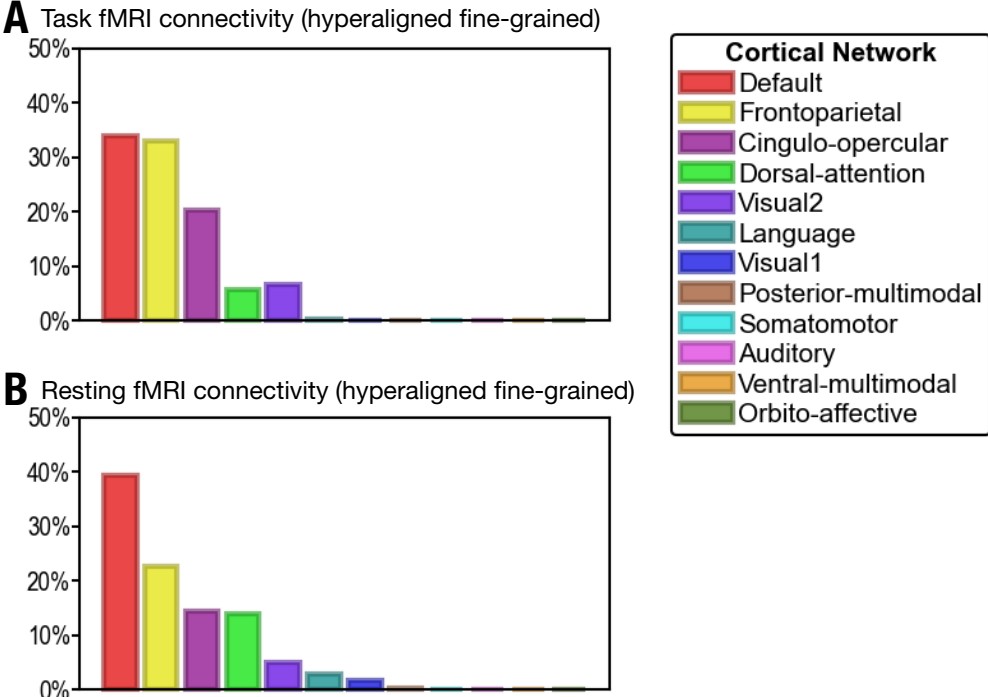

**Appendix 1—figure 11.** Proportion of vertices of the 30 most predictive regions in 12 cortical networks defined in *Ji et al., 2019*. (**A**) The 30 regions that are most predictive of general intelligence based on hyperaligned fine-grained task functional magnetic resonance imaging (fMRI) connectivity are in the default (34.0% of vertices in the 30 regions), frontoparietal (33.0%), cingulo-opercular (20.2%), dorsal attention (5.7%), and visual 2 (6.7%) networks. (**B**) The 30 regions that are most predictive of general intelligence based on hyperaligned fine-grained resting fMRI connectivity are in the default (39.3%), frontoparietal (22.5%), cingulo-opercular (14.4%), dorsal attention (13.9%), visual 2 (4.9%), language (2.8%), and visual 1 (1.7%) networks. The 12 cortical networks are defined based on *Ji et al., 2019*. Note that different cortical network parcellations are in agreement with each other in general, and the proportion of cortical vertices in these 12 networks is similar to the proportion in the seven cortical systems based on *Yeo et al., 2011*. Most of the vertices are in default, frontoparietal, and dorsal attention networks based on both *Yeo et al., 2011* and *Ji et al., 2019*; some regions in the ventral attention network based on *Yeo et al., 2011* are labeled as cingulo-opercular in *Ji et al., 2019*.

## Assessing the effects of head motion

The level of in-scanner head motion (usually measured as framewise displacement [FD]) is correlated with many subject measures, and participants who move more in the scanner are more likely to have lower cognitive test scores (*Siegel et al., 2017*). Therefore, when the connectivity pattern is affected by motion-related artifacts, it is possible for prediction models to take advantage of these spurious patterns and predict these subject measures to some extent. For the dataset used in this study, the correlation between $g$ and FD is $r(874) = -0.29$, $p < 10^{-18}$, which means a perfect head motion predictor can account for 8.6% of variance in general intelligence (*Appendix 1—figure 12A*). To assess the influence of head motion on our prediction models, we performed two additional analyses.

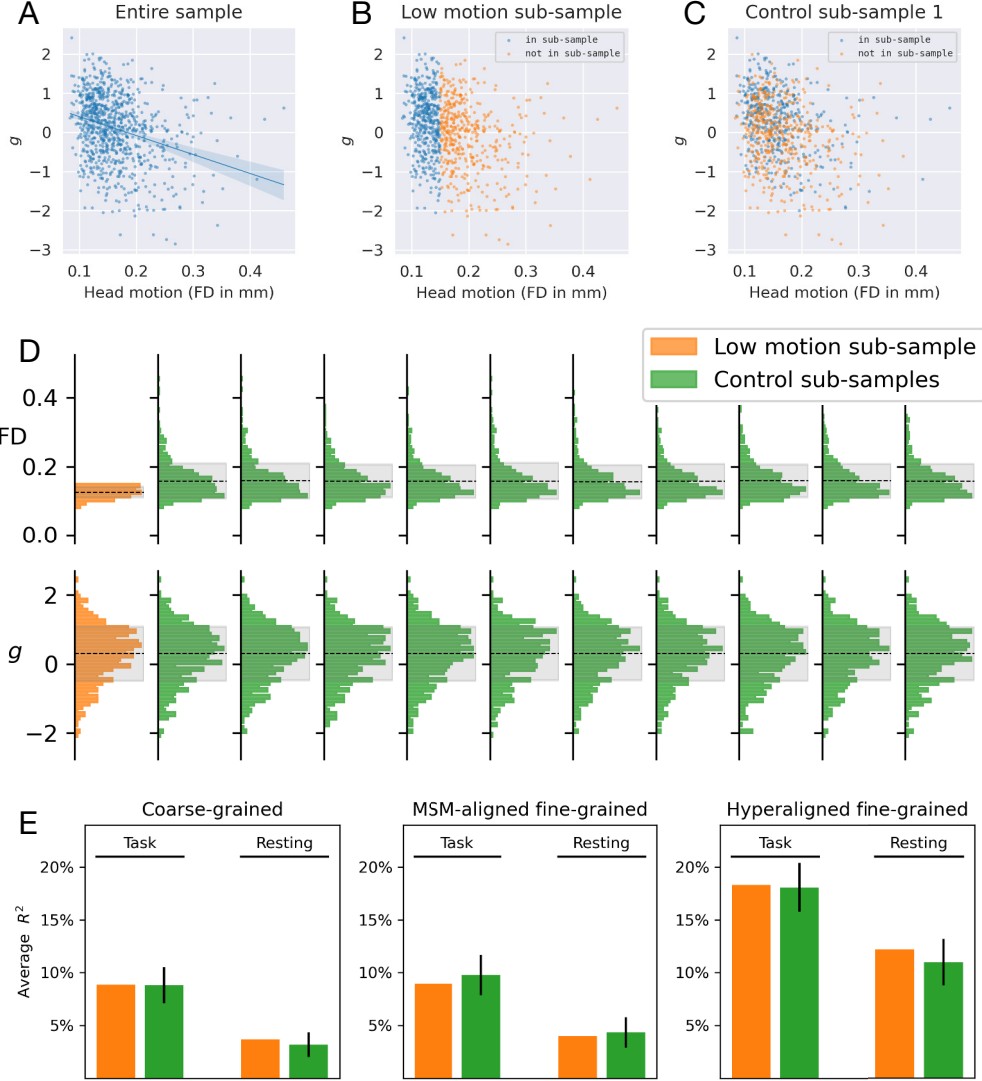

**Appendix 1—figure 12.** Prediction performance based on a low head motion sub-sample. General intelligence has a moderate correlation with head motion (**A**), and a 'motion detector' can predict general intelligence to some extent. To assess the effect of head motion variation on prediction performance, we trained another set of models based on a sub-sample of participants with minimal head motion (framewise displacement <0.15 mm, n = 437, **B**), and another 10 sets of models based on control sub-samples (see **C** for an example control sub-sample). The control sub-samples had similar general intelligence variation but larger head motion variation compared with the low-motion sub-sample (**D**). Prediction performance was similar between the low-motion sub-sample and the control sub-samples (**E**), suggesting larger variation in head motion level (while keeping sample size constant) has little effect on the prediction performance for general intelligence.

In the first analysis, we created a low-motion sub-sample of the dataset and evaluated prediction performance based on this sub-sample. This sub-sample comprises participants whose average FD is less than 0.15 mm across all fMRI runs and has a sample size of 437. Because sample size has a great influence on prediction performance (*Appendix 1—figure 8*), we created 10 control sub-samples. These control sub-samples have the same sample size and almost identical mean and standard deviation of *g* as the low-motion sub-sample (*Appendix 1—figure 12*D), but have the full spectrum of head motion (C). Therefore, if the prediction models take advantage of the correlation between *g* and head motion to predict *g*, we expect the model performance based on the control sub-samples to outperform the low-motion sub-sample.

Across all three kinds of connectivity profiles (coarse-grained, MSM-aligned fine-grained, hyperaligned fine-grained) and two kinds of fMRI data (task fMRI, resting fMRI), we observed no difference in prediction performance between models based on the low-motion sub-sample and models based on control sub-samples (all $p > 0.11$; *Appendix 1—figure 12*E). This suggests that head motion has little influence on the performance of our prediction models.

Note that in the low-motion sub-sample, small variation in head motion still exists (*Appendix 1—figure 12*D) and is slightly correlated with $g$ ($r = -0.14$). Therefore, it is possible for a prediction model to take advantage of motion-related connectivity patterns and account for up to 1.9% of variance in $g$ if it can detect motion from functional connectivity perfectly. To further assess the effect of head motion, we performed a second analysis in a more strict way.

In the second analysis, we regressed out FD from functional connectivity profiles and used the residual profiles to build our prediction models. This allows us to completely remove the variance in functional connectivity that covariates with head motion. Traditionally certain preprocessing pipelines are used to mitigate the effects of head motion on functional connectivity and reduce the prediction performance overestimation caused by head motion (*Siegel et al., 2017*; *Ciric et al., 2017*; *Ciric et al., 2018*). However, it is unclear which preprocessing pipeline is most effective in removing motion-related artifacts in fine-grained connectivity and it is practically challenging to compute fine-grained functional connectivity using multiple preprocessing pipelines. Therefore, we chose to use regression to remove covariation between functional connectivity and head motion. Note that regression is an aggressive approach and will also remove variance in 'true' functional connectivity that covariates with FD. As a result, the maximal possible $R^2$ is no longer 100% but rather 91.4% because the 8.6% of variance in $g$ that covaries with FD can never be accounted for by these residual profiles.

To account for the effect of the new $R^2$ ceiling, we trained another 100 sets of prediction models, where each set was based on functional connectivity residuals after regressing out a random variable that had the same level of correlation with $g$ (i.e., $r = 0.29$), and therefore had the same $R^2$ ceiling. For simplicity, we will refer to these models as control models and the models based on regressing out FD as FD-residual models. With these models, it is possible to separate the effect of $R^2$ ceiling by comparing original models and control models, and the effect specific to head motion by comparing control models with FD-residual models.

Compared with the original models, control models had lower $R^2$, and the difference was 2.8%, 3.1%, and 4.2% for coarse-grained, MSM-aligned fine-grained, and hyperaligned fine-grained connectivity profiles respectively based on task fMRI data; the difference was 1.9%, 2.5%, and 3.9% respectively based on resting fMRI data. This demonstrates that regressing out variables correlating with $g$ from functional connectivity profiles will reduce model performance in general. Note that the $R^2$ difference was larger for connectivity profiles that are more predictive of $g$ (e.g., hyperaligned fine-grained connectivity profiles for both task and resting fMRI data), suggesting that high-performance models are more influenced by the $R^2$ ceiling, probably because their performances are closer to the ceiling.

FD-residual models had lower $R^2$ compared with these control models, and the difference was 2.5%, 2.8%, and 2.7% respectively based on task fMRI data, and 1.3%, 2.0%, and 1.1% respectively based on resting fMRI data (*Appendix 1—figure 13*). These results suggest that head motion has a small (1.1–2.8%) but statistically significant effect on predicting $g$ based on functional connectivity. Based on resting fMRI data, the effect of head motion was smaller for fine-grained connectivity profiles that are hyperaligned compared with those MSM-aligned ($\Delta R^2 = 0.9\%$, $p=0.01$). This suggests that using functional connectivity based on hyperaligned data may reduce the influence of head motion on prediction models.

## Task fMRI functional connectivity

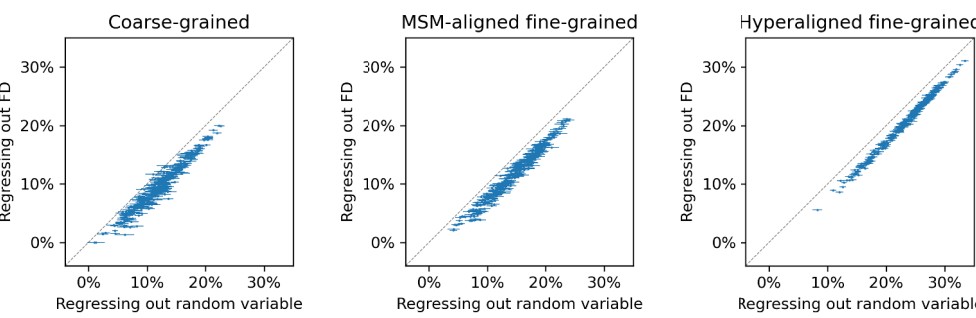

## Resting fMRI functional connectivity

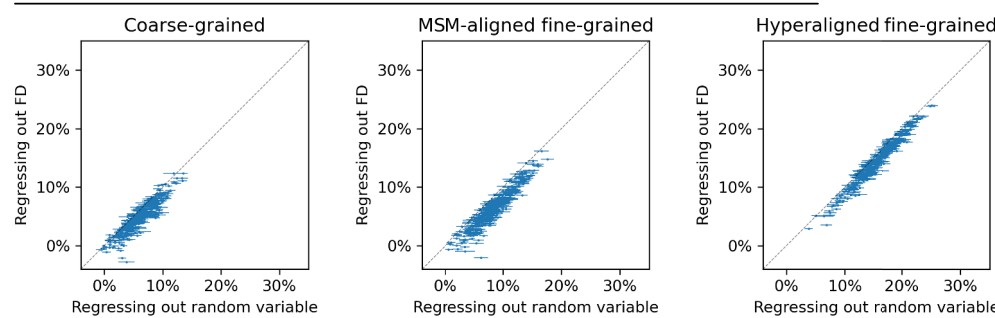

**Appendix 1—figure 13.** Regressing out head motion from functional connectivity. We regressed out head motion (measured as framewise displacement) from functional connectivity profiles and built prediction models based on the residual connectivity profiles, and compared their performance with 100 sets of control models. Each dot is a brain region, and error bars denote the standard deviation across 100 control model sets. For each set of control models, a random variable that has the same level of correlation with $g$ was regressed out from functional connectivity profiles instead of head motion to control for the effect of lower $R^2$ ceiling caused by regression. Compared with the control models, models based on regressing out head motion had slightly lower performance (range: 1.1–2.8%), as demonstrated by that the dots are slightly below the diagonal in general. This difference suggests that the shared variance between functional connectivity and head motion may cause a slight overestimation of model performance. However, the effect is small and cannot explain the difference between models based on different types of connectivity profiles.

