## [Decision Letter]

**Acceptance summary:**

In this work, Feilong and colleagues investigated the degree to which the strength of functional connectivity is predictive of general intelligence, and the degree to which that predictive power is improved using hyperalignment procedures. More specifically, the authors showed a two-fold increase in variance explained in general intelligence when using fine-grained hyperaligned connectivity compared with coarse-grained hyperaligned connectivity. This is a very clearly written paper that presents an important result, which has the potential of great impact on the field of behavioral prediction.

**Decision letter after peer review:**

Thank you for submitting your article "The neural basis of intelligence in fine-grained cortical topographies" for consideration by *eLife*. Your article has been reviewed by three peer reviewers, one of whom is a member of our Board of Reviewing Editors, and the evaluation has been overseen by Floris de Lange as the Senior Editor. The following individuals involved in review of your submission have agreed to reveal their identity: Janine Diane Bijsterbosch (Reviewer #2); Evan Gordon (Reviewer #3).

The reviewers have discussed the reviews with one another and the Reviewing Editor has drafted this decision to help you prepare a revised submission.

Summary:

In this work, Feilong and colleagues use the Human Connectome Project fMRI data to investigate the degree to which the strength of functional connectivity is predictive of general intelligence, and the degree to which that predictive power is improved using the hyperalignment procedures their lab has developed. More specifically, the authors predict general intelligence using either coarse-grained functional connectivity (based on 360 ROIs) or fine-grained functional connectivity (vertex-wise) after hyperalignment. The results show a two-fold increase in variance explained in general intelligence between coarse-grained and fine-grained connectivity. This is a very clearly-written paper that presents an important result, which has the potential of great impact on the field of behavioral prediction. However, the reviewers and editors do have some significant concerns with the predictive modeling presented in this work.

Essential revisions:

1) A major contribution of this study is the massive improvement in prediction performance using connectivity hyperalignment and fine-grained functional connectivity. As such it is important that code for this study be made publicly available, so that other researchers can test and replicate the authors' analyses under new conditions and datasets. Our understanding is that the connectivity hyperalignment code from the previous study (Guntupalli et al., 2018) is available in PyMVPA. However, our experience is that the code is not easy to use. As such, we believe it is important that the code specific to this study be made publicly available. More specifically, code utilized in this study to apply the existing connectivity hyperalignment code to the HCP dataset should be made available. Furthermore, code for computing fine-grained functional connectivity together with PCA+ridge regression and nested cross-validation should also be made available.

2) With regards to the leave-one-family-out nested cross-validation procedure, previous studies (e.g., Varoquaux et al., 2017) have suggested a single round of cross-validation can be sensitive to the particular split of the data. A more robust procedure would be to perform 10-fold nested cross-validation procedure 50 times. The prediction performance is then averaged across the 50 x 10 = 500 folds. In the case of the HCP data, care should be taken to handle family structure, i.e., within a single 10-fold nested cross-validation procedure, a family should not be split across the 10 folds. We believe that such a procedure is especially important for this study because the major result here is the huge improvement in prediction performance.

3) The authors should clarify what hyperparameters were tuned in their nested cross-validation. Our understanding is that the authors tune the number of PCA components and ridge regularization parameter. However, hyperalignment has a few hyperparameters as well. Did the authors use the exact same hyperalignment parameters as their previous studies? If so, this should be clearly stated in this study. If different hyperalignment parameters were used, then these hyperalignment hyperparameters should also be tuned within the nested cross-validation framework.

4) The residuals of fine-grained connectivity profiles were obtained after subtracting coarse-grain connectivity. Why was subtraction used here, rather than regressing out (i.e., orthogonalizing with respect to) the coarse-grained connectivity?

5) The authors have generally done a good job controlling for motion-related confounds, which can be a serious issue in the HCP data. In fact, Siegel et al., 2016, demonstrated that many behavioral measures, including intelligence, appeared to be spuriously related to motion effects. This is a particular concern for predictive modeling of the type done in the current work, as it is never clear when predictions are being made based on real aspects of the data vs. when predictions are being made based on intelligence-correlated motion artifact. However, the authors did not "scrub" their data (completely remove high-motion frames), as Siegel et al. did. This could be an issue, as Siegel et al. appeared to show that scrubbing by itself could remove a good portion of spurious behavioral covariance, and Ciric et al., 2017, showed that scrubbing removes different portions of the motion-related artifact than nuisance regression of the type performed by the authors does. Have the authors tested whether their strong FC-behavior predictive power survives more stringent removal of motion frames?

6) Glasser et al., 2016, showed that machine learning approaches could generate individual-specific versions of their parcellation in HCP data that were substantially variable across subjects (even after MSM alignment). This is, of course, a different approach to hyperalignment, at the parcel level rather than the fine-grained vertex level. Have the authors considered testing whether hyperalignment results in better predictive power than such individualized parcel estimates?

7) How does the bootstrapping handle the family structure in the data? If family structure is not taken into account, the authors should justify why that is the case.

8) The Materials and methods states that a linear regression model was used to control for area size in dissimilarity estimates. It would be useful to provide more details here please? For example, is the model fit across subjects or across regions within subject?

9) Some more details about the implementation of permutation testing of the model would be helpful. For example, was each model fully re-trained in each permutation, including the parameter optimization?

10) The fine-grained functional connectivity has richer features than coarse-grained, leading to higher dimensionality in the PCA step (Figure 3—figure supplement 5). We wonder if this might contribute to improved prediction accuracy. Related to this, it appears that there may also be a relationship between PCA dimensionality and regularization parameter, such that more regularization may be needed when more PCs are used in the model. It would be interesting to test the effect of fixing the PCA dimensionality (and perhaps also the regularization) across all models to control model complexity.

---

## [Author Response]

Essential revisions:1) A major contribution of this study is the massive improvement in prediction performance using connectivity hyperalignment and fine-grained functional connectivity. As such it is important that code for this study be made publicly available, so that other researchers can test and replicate the authors' analyses under new conditions and datasets. Our understanding is that the connectivity hyperalignment code from the previous study (Guntupalli et al., 2018) is available in PyMVPA. However, our experience is that the code is not easy to use. As such, we believe it is important that the code specific to this study be made publicly available. More specifically, code utilized in this study to apply the existing connectivity hyperalignment code to the HCP dataset should be made available. Furthermore, code for computing fine-grained functional connectivity together with PCA+ridge regression and nested cross-validation should also be made available.

We thank the reviewers for bringing this up. We do plan to release the code and the derived data in full. We have prepared the code for prediction and cross-validation, as well as the data used in these steps, so that they will be released as soon as the final revision is accepted for publication. Researchers can use these resources to build their own prediction models and study other measures of interest.

Execution of the code for computing fine-grained functional connectivity together with performing PCA on functional connectivity usually requires a high-performance computing cluster due to the size of the dataset and the nature of fine-grained functional connectivity. We will also release the code for these steps to maximize the replicability, but the precomputed data is recommended for most users.

All code will be published openly, and derived data of the HCP dataset will be released according to the Data Use Terms of the dataset (https://www.humanconnectome.org/study/hcp-young-adult/document/wu-minn-hcp-consortium-open-access-data-use-terms).

2) With regards to the leave-one-family-out nested cross-validation procedure, previous studies (e.g., Varoquaux et al., 2017) have suggested a single round of cross-validation can be sensitive to the particular split of the data. A more robust procedure would be to perform 10-fold nested cross-validation procedure 50 times. The prediction performance is then averaged across the 50 x 10 = 500 folds. In the case of the HCP data, care should be taken to handle family structure, i.e., within a single 10-fold nested cross-validation procedure, a family should not be split across the 10 folds. We believe that such a procedure is especially important for this study because the major result here is the huge improvement in prediction performance.

We repeated our analysis using 50 repetitions of 10-fold cross-validation, and found very similar results as leave-one-family-out cross-validation. The small difference can be explained by the difference in training data size (90% vs. 99.8% of the dataset). We summarized these results (along with results based on 2-fold, 3-fold, and 5-fold) in Figure 3—figure supplement 8.

The issue with leave-one-trial-out cross-validation discussed in Varoquaux et al., 2017, was because trials from the same run were split between training and testing data during leave-one-trial-out cross-validation, even though they are not independent. We assessed a similar issue in the individual differences context, that is, allowing members from the same family to be split between training and testing data. We found a consistent overestimation of prediction performance when the family structure was not controlled. We added these results as Figure 3—figure supplement 9.

3) The authors should clarify what hyperparameters were tuned in their nested cross-validation. Our understanding is that the authors tune the number of PCA components and ridge regularization parameter. However, hyperalignment has a few hyperparameters as well. Did the authors use the exact same hyperalignment parameters as their previous studies? If so, this should be clearly stated in this study. If different hyperalignment parameters were used, then these hyperalignment hyperparameters should also be tuned within the nested cross-validation framework.

We used the same hyperparameters as in our previous studies. Specifically, we used the solution to the orthogonal Procrustes problem to derive the transformation matrices (i.e., an improper rotation in a high-dimensional space), which translates to “reflection=True, scaling=False” in PyMVPA terms. Other parameters are identical to the default parameters of PyMVPA's “Hyperalignment” class, except for that “zscore_common” was turned off because the input was connectivity rather than zscored response time series. We have expanded the corresponding subsection in the Materials and methods to explain the algorithm and parameter choices:

“In this study, hyperalignment was performed for each brain region separately (i.e., in a similar manner to Haxby et al., 2011 rather than Guntupalli et al., 2016), and the transformation was constrained to be an improper rotation (i.e., rotation that allows reflection) with no scaling. […] For example, an improvement of prediction performance after searchlight hyperalignment can be caused by better alignment of information in a region, but it can also be caused by additional information moved into the region or noise moved out of the region.”

4) The residuals of fine-grained connectivity profiles were obtained after subtracting coarse-grain connectivity. Why was subtraction used here, rather than regressing out (i.e., orthogonalizing with respect to) the coarse-grained connectivity?

In this special case, the residual obtained by subtraction is equivalent to the residual obtained by regression. We have added the following text to the Materials and methods to clarify this detail:

“Because a coarse-grained connectivity profile is comprised of region-by-region averages of the corresponding fine-grained connectivity profile, a regression model using coarse-grained connectivity profile (expanded to the same size as the fine-grained connectivity profile) as the independent variable and fine-grained connectivity profile as the dependent variable will always have a slope of 1. Therefore, the difference between the two profiles is also the residual of the regression model.”

5) The authors have generally done a good job controlling for motion-related confounds, which can be a serious issue in the HCP data. In fact, Siegel et al., 2016, demonstrated that many behavioral measures, including intelligence, appeared to be spuriously related to motion effects. This is a particular concern for predictive modeling of the type done in the current work, as it is never clear when predictions are being made based on real aspects of the data vs. when predictions are being made based on intelligence-correlated motion artifact. However, the authors did not "scrub" their data (completely remove high-motion frames), as Siegel et al. did. This could be an issue, as Siegel et al. appeared to show that scrubbing by itself could remove a good portion of spurious behavioral covariance, and Ciric et al., 2017, showed that scrubbing removes different portions of the motion-related artifact than nuisance regression of the type performed by the authors does. Have the authors tested whether their strong FC-behavior predictive power survives more stringent removal of motion frames?

We performed two additional analyses to assess the effect of motion on our prediction models, and added a new subsection “Assessing the effects of head motion” and two supplementary figures (Figure 3—figure supplement 12 and Figure 3—figure supplement 13).

In the first analysis, we created a low motion sub-sample (average FD < 0.15 mm) and compared prediction performance based on this sub-sample with 10 control sub-samples. Each control sub-sample had the same sample size and a similar distribution of intelligence as the low motion sub-sample, but the distribution of head motion was more similar to the entire sample (i.e., including large motion participants). Prediction performance was similar between the low motion sub-sample and the control sub-samples (Figure 3—figure supplement 12).

In the second analysis, we completely regressed out head motion (measured as FD) from the connectivity profiles and built the prediction models based on the residuals. Compared with the baseline (regressing out a random variable correlated with *g*), by regressing out FD from functional connectivity the average model performance across 360 regions dropped by 1.1% to 2.8% for different kinds of connectivity profiles (Figure 3—figure supplement 13).

Together these results demonstrate that functional connectivity patterns correlated with head motion may cause a small overestimation of model performance. However, this effect is small (Δ*R^2^* = 1.1% to 2.8%) for our models and doesn't explain the large difference between different connectivity profile types.

6) Glasser et al., 2016, showed that machine learning approaches could generate individual-specific versions of their parcellation in HCP data that were substantially variable across subjects (even after MSM alignment). This is, of course, a different approach to hyperalignment, at the parcel level rather than the fine-grained vertex level. Have the authors considered testing whether hyperalignment results in better predictive power than such individualized parcel estimates?

Hyperalignment and individualized parcellations both can resolve idiosyncratic region boundaries on the anatomy (e.g., Jiahui et al., 2020). In this work (Figure 3—figure supplement 2) and our previous work (Feilong et al., 2018) we found that hyperalignment benefits individual differences analysis at both coarse and fine spatial scales, but the effect was much larger for fine-scale information than coarse-scale information. A very recent work, Kong et al., 2021, shows that individual-specific parcellations also improve prediction performance. Our prediction performance for fluid intelligence (Figure 3—figure supplement 6A) seems to surpass Kong et al., 2021 (Figure 11). However, many methodological details differ and a systematic comparison in future work is more appropriate for conclusions. We have added discussions on alternative approaches for aligning coarse-scale information:

“Hyperalignment resolves functional topographic idiosyncrasies at both coarse and fine spatial scales (see Jiahui et al., 2020, for an example of aligning functional regions), which is critical for assessing individual differences in information processing. […] However, the improvement for coarse-grained functional connectivity was smaller than for fine-grained functional connectivity in predicting general intelligence (Figure 3—figure supplement 2).”

7) How does the bootstrapping handle the family structure in the data? If family structure is not taken into account, the authors should justify why that is the case.

We ensured that the training data didn't contain participants from the same family as the test participant. Specifically, our bootstrapping procedure only resamples participants used for model evaluation, so that the sampling distribution is representative of the original training data amount. We have added a more detailed description of this procedure:

“The bootstrapping procedure used here only affects the participants used for model evaluation (i.e., the test set), and the model used to predict a participant's score was always trained with the same training set (i.e., participants who are not from the same family as the test participant). […] Therefore, we only bootstrapped the participants used for model evaluation, so that the sampling distribution is not biased by training sample size or dependency.”

8) The Materials and methods states that a linear regression model was used to control for area size in dissimilarity estimates. It would be useful to provide more details here please? For example, is the model fit across subjects or across regions within subject?

We averaged the matrix norm across all participant pairs for each region, and used the average in the regression model, so it is a single regression model across 360 brain regions. We have rewritten the description to clarify the details:

“We measured the level of each region's functional topographic idiosyncrasy as the average dissimilarity of hyperalignment transformation matrices across participant pairs for that region, after correcting for region size. […] We used the residual of the linear regression model to depict the heterogeneity of a region's functional topography across individuals.”

9) Some more details about the implementation of permutation testing of the model would be helpful. For example, was each model fully re-trained in each permutation, including the parameter optimization?

In each permutation, we permuted general intelligence scores using multi-level block permutation, and used these permuted scores for the entire process, including parameter optimization, model training, prediction, and evaluation. We have added the following text in the methods description to make it clearer:

“Each time, we permuted general intelligence scores across the entire dataset in the beginning, and re-ran the entire prediction pipeline with these permuted scores. […] In other words, we repeated the entire process – including parameter optimization, training, prediction, and model evaluation – using permuted general intelligence score as the target variable instead of the original general intelligence score.”

10) The fine-grained functional connectivity has richer features than coarse-grained, leading to higher dimensionality in the PCA step (Figure 3—figure supplement 5). We wonder if this might contribute to improved prediction accuracy. Related to this, it appears that there may also be a relationship between PCA dimensionality and regularization parameter, such that more regularization may be needed when more PCs are used in the model. It would be interesting to test the effect of fixing the PCA dimensionality (and perhaps also the regularization) across all models to control model complexity.

We trained another 8 sets of prediction models, each with a fixed hyperparameter combination, to assess the effect of hyperparameter choices. These 8 combinations are 4 levels of dimensionality reduction (80 PCs, 160 PCs, 320 PCs, or all PCs) × 2 levels of regularization (α = 0.1 or α = 10^-20^) that were most frequently chosen by nested cross-validation (Figure 3—figure supplement 5). We found that reducing the number of PCs reduced model performance for both coarse- and fine-grained connectivity profiles (except for overfit models due to insufficient regularization). However, even with the same number of PCs, prediction models based on hyperaligned fine-grained connectivity profiles still had a great advantage over those based on MSM-aligned fine-grained and coarse-grained connectivity profiles. We have added a new supplementary figure (Figure 3—figure supplement 10) to demonstrate the effect of hyperparameter choices.